# Dynamic allele usage of X-linked genes ameliorates neurodevelopmental disease phenotypes in brain organoids

M. Bertin[1,19], H. Todorov [1,19], S. Frank [2,19], S. Käseberg [1], R. Menon[2,3], E. Gabassi [2], C. Foerster[1,4], N. Bobon[5], F. Furlanetto [2], A. Soliman [1], H. M. B. Ibrahim [1], V. Engelhardt [1], L. Birschmann [2], H. Brennenstuhl [6,7], B. Lohrer[2], A. Mas-Sanchez[1], E. Cesare[8], J. Winter[1], J. Krummeich[1], J. Winkler [9], B. Winner [10,11], E. Weis[1], S. Diederich[1], K. Luck[12], P. Lunt[13], S. Gerber [1], P. Baumann[5,12,14], N. Elvassore [8], B. Berninger [3,15,16,17,18], MF Basilicata [1,12], S. Schweiger[1,4,12,18] ✉, S. Falk [2] ✉ & M. Karow [2] ✉

While random X-chromosome inactivation in female cells of placental mammals silences one allele of the majority of X-chromosomal genes, a considerable fraction is only incompletely and variably inactivated. Human model systems to study the dynamics of incomplete X-inactivation are limited mostly to postmortem tissue, thereby disregarding developmental trajectories. Here, we used clonal human female induced pluripotent stem cells to track allele-specific expression of X-chromosomal genes along neural differentiation. We discovered dynamic reactivation and late-silencing of gene expression from the inactive X-chromosome leading to differentiation-induced locus- and lineage-specific usage of the two X-chromosomal alleles. In brain organoids modeling Opitz BBB/G syndrome, an X-linked neurodevelopmental disorder, reactivation of alleles from the inactive X-chromosome rescued cellular phenotypes and led to intermediate manifestations in female tissue. Taken together, our data demonstrate that alleles on the inactive X-chromosome can serve as a critical reservoir dynamically used during differentiation, thereby enhancing resilience of female neural tissue.

In somatic cells of female eutherian mammals, one of the two X-chromosomes is randomly inactivated early during embryonic development leading to a silent gene reservoir[1–3]. However, a significant proportion of genes on the inactive X-chromosome have been found to escape the inactivation process[4–6]. In addition to constitutive escapees, which escape X-chromosome inactivation (XCI) in all cells and tissues, a similar number of genes escape inactivation facultatively. Facultative escape entails biallelic expression that is observed only in selected tissues and is heterogenous between individuals[7–9]. Both XCI as well as escape show considerable differences in timing and underlying mechanisms between eutherian mammals[2,10–12], pointing

towards distinct regulatory principles and highlighting the importance of species-specific model systems. Escape from XCI results in biallelic expression and frequently in a female expression bias, i.e., higher expression levels of the same gene in female compared to the corresponding male tissue[8,13]. However, there is little experimental data interrogating the developmental timeline of XCI, escape, and the establishment of facultative escape in human cells. Taking advantage of the clonal nature and the retention of the X-chromosome activation status in human induced pluripotent stem cells (iPSCs)[14], we here monitored XCI and escape throughout neural differentiation from a defined point-of-origin. We observed dynamic reactivation and late-

silencing of alleles on the inactive X-chromosome induced by differentiation. Furthermore, we used publicly available single cell RNA sequencing data from spinal cord of human embryos at two different developmental stages to confirm a dynamic time- and cell-type specific usage of gene expression from the inactive X-chromosome in vivo. Strikingly, many of these reactivating genes are linked to neurodevelopmental disorders (NDDs).

NDDs exhibit an intriguing sex bias with females being less often and less severely affected than males[15]. While NDD associated genes are evenly distributed across all autosomes, they are overrepresented on the X-chromosome[16], highlighting the particularly important role of the X-chromosome in the phenotype formation of NDDs. In males, most X-linked genes are hemizygous, i.e., males only carry the allele inherited from the mother. In contrast, random XCI in female cells results in a mosaic expression of alleles from the two X-chromosomes in female tissues, substantially influencing phenotype development of X-linked gene defects. However, hemizygosity of X-linked genes in males and random XCI in females only partially explain why females are disproportionately less affected by NDDs[17–19]. We hypothesized that dynamic X-chromosome reactivation provides a protective mechanism in female neural tissue, reducing the frequency and severity of NDDs in females. To test this hypothesis, we employed an iPSC-based brain organoid model of Opitz BBB/G syndrome (OS), a clinically variable NDD characterized by cerebellar vermis hypoplasia, microcephaly, ventral midline aberration, developmental delay, and intellectual disability[20–23]. OS is caused by mutations in the X-linked *MID1* gene, which we found dynamically reactivated during neural differentiation. While brain organoids derived from iPSC of a male OS patient exhibited a dramatic decrease of differentiated neurons accompanied by a delayed cell cycle exit of neural stem and progenitor cells, we found that organoids derived from female cells with the same mutation on their active X-chromosome showed a markedly milder, intermediate phenotype. To interrogate the influence of the allele on the inactive X-chromosome, we employed CRISPR/Cas9 to introduce the same patient-specific loss-of-function mutation on both the active and inactive X-chromosome. Strikingly, brain organoids derived from these homozygous OS female iPSC lines exhibited a phenotype similar to the hemizygous male organoids. This demonstrates that the allele on the inactive X-chromosome substantially influences phenotype development.

Taken together, we here dissected the temporal usage of the two X-chromosomes along neural differentiation trajectories in brain organoids and human embryos and thereby uncloaked dynamic reactivation and late silencing of individual alleles on the inactive X-chromosome as a mechanistic basis for facultative escape. Furthermore, we showed that this fine-tuned symphony of dynamic biallelic X-chromosomal gene expression has substantial protective effects rescuing cellular phenotypes of OS and contributes to resilience of female neural tissue.

## Results

### Reactivation of X-linked genes during neural differentiation

To trace the X-activation status along differentiation trajectories, we took advantage of the clonal nature of human iPSCs which inactivate the same X-chromosome as the cell of origin[14]. Female fibroblasts were reprogrammed into iPSCs and upon clonal selection we determined which X-chromosome is active in which clone by using Quantification of Allele-Specific Expression by Pyrosequencing (QUASEP) assays (Fig. S1a–d, Supplementary Data 1). This technique measures allelic expression of selected genes based on heterozygous single nucleotide polymorphisms (SNPs) and enables distinguishing the two X-chromosomes. Establishing this defined point-of-origin in iPSCs allowed us to follow individual allele usage along the differentiation axis. Using bulk RNA-sequencing of iPSCs, neural stem and progenitor cells (NPCs), and neurons (Fig. 1a) of three distinct control (ctrl) cell lines from three different donors (M-ctrl, J-ctrl and A-ctrl, Fig. S1a and k,

Supplementary Data 1), we performed a global characterization of the allele usage of X-linked genes leveraging the presence of expressed heterozygous sequence variants covered by at least 20 reads (see Methods). Principal component analysis of X-chromosomal allelic expression profiles showed a clear distinction between iPSCs, NPCs, and neurons (Fig. S1e). Surprisingly, we identified a set of genes that were monoallelically expressed in iPSCs but exhibited differentiation-dependent reactivation (reactivating genes) resulting in biallelic expression in either NPCs, neurons, or both cell types (Fig. 1b, Fig. S1f). Although this allele usage analysis depends on the presence of expressed heterozygous sequence variants that vary between donors and cell lines, still a striking number of 22 genes were reactivated in at least two of the cell lines used. Furthermore, we detected full escapees that were expressed biallelically in iPSCs, NPCs, and neurons with 35 genes found escaping in at least two cell lines (Fig. 1c, Fig. S1g). Finally, a group of genes (5 in at least two of the cell lines) switched from biallelic expression in iPSCs to monoallelic expression in NPCs and neurons (late-silenced genes, Fig. 1d, Fig. S1h). To ensure that detection of biallelic expression is independent of the overall expression level of a gene, we correlated mRNA-expression (transcripts per million) with allelic expression (Xi/total allelic expression) for all genes together (Fig. S1i) and for individual genes (Fig. S1j). Importantly, reactivated and late-silenced genes showed no significant correlation between overall gene expression levels and allelic expression, confirming that mono-allelic expression for reactivated and late-silencing genes was not spuriously detected due to lack of overall gene expression in the respective cell types (Fig. 1b, c, d heatmaps and Fig. S1i, j). To validate the dynamic allelic usage from the inactive X-chromosome, we employed QUASEP assays. While *CASB*, a gene known to constitutively escape XCI[24], was expressed biallelically in all tested cells, including iPSCs, *ZNF185* showed purely monoallelic expression throughout neuronal differentiation (Fig. S2a, b). In contrast *MID1* and *GPM6B*, two genes located in close vicinity on the short arm of the X-chromosome, were reactivated in NPCs and neurons indicating locus-specific reactivation (Fig. S2c–g). These data confirmed that, apart from purely monoallelic genes and full escapees, additional categories of X-chromosomal genes exist with dynamic expression from the inactive X-chromosome during neural differentiation thereby resulting in facultative escape in our model.

Loss of the lncRNA *XIST* and consequently erosion of X-inactivation is a known phenomenon in iPSCs occurring mainly in higher cell culture passages[25–27]. In the bulk RNA-sequencing analysis, the A-ctrl line retained high *XIST* levels, however the M-ctrl and J-ctrl lines were low *XIST* expressing (Fig. S3a). To unambiguously visualize differentiation-induced reactivation of X-chromosomal genes in *XIST* expressing NPCs at single cell resolution, we employed RNAscope, detecting unspliced pre-mRNA of selected target genes together with an *XIST* cloud surrounding the inactive X-chromosome. H3K27 trimethylation and *XIST* enrichment on the inactive X-chromosome were continuously monitored in all iPSC and NPC samples used in the experiments (Fig. S3b–e), with *XIST* high expressing lines showing localized *XIST* expression in more than 70% of the cells. No signals were seen in negative controls where no probes were used, and in samples pre-treated with RNase before probe incubation (Fig. S3f). Visualizing the unspliced pre-mRNA allows to detect the origin of the nascent RNA and hence to detect whether a given transcript is expressed monoallelically (one spot) or biallelically (two spots, Fig. 1e). The inactive *SPIN3* gene showed monoallelic expression with no reactivation of expression from the inactive X-chromosome in iPSCs or NPCs (Fig. 1f, g) while the full-escapee *NLGN4X* was biallelically expressed in 40–50% of both, iPSCs ad NPCs (Fig. 1h, i). The reactivated genes *MID1* and *GPM6B*, however, showed only one *pre-mRNA* spot distant to the *XIST* positive territory in the vast majority of iPSC nuclei in all cell lines tested (Fig. 1j, k, Fig. S3g–l). In NPCs, however, we observed a pronounced increase of *XIST* positive nuclei showing two sites of nascent

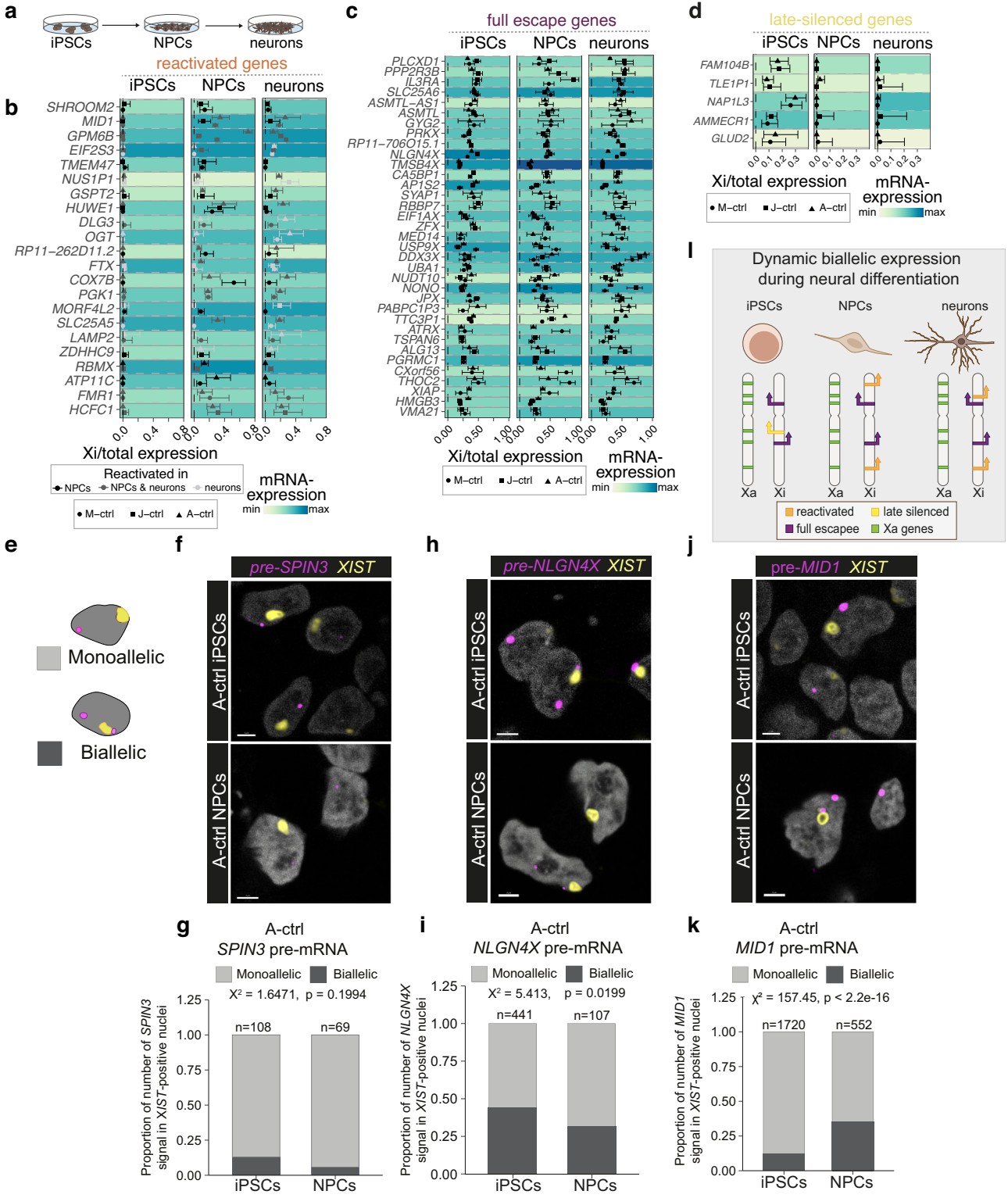

RNA transcription, with one site co-localizing with the *XIST* signal (Fig. 1j, k, Fig. S3g–l). These data show on a single cell level that the two genes *MID1* and *GPM6B* switch from monoallelic expression in iPSCs to biallelic expression upon differentiation into NPCs in non-eroded, *XIST* expressing cells (findings summarized in Fig. 1l).

## Re-activating genes cluster and are associated with active chromatin

We next overlapped the different types of biallelic genes we detected with constitutive and facultative escapees that were reported in

previous studies[8,13]. This analysis revealed that reactivated, escape and late-silenced genes shared a significantly higher overlap than expected by chance with previously reported escapees in humans (Fig. S3m, Supplementary Data 2). Biallelic expression of X-chromosomal genes often leads to higher expression levels in female cells. Therefore, we next assessed whether biallelically expressed genes can show a female expression bias by looking at the female to male ratios of mRNA-expression across 44 human tissues from the GTEx project[13]. We found a marked increase in the proportion of genes with a female expression bias in multiple tissues not only for full-escapees but also for

**Fig. 1 | Reactivation of X-linked genes during neural differentiation.**
**a** Experimental scheme depicting neural 2D differentiations. Scatter plots showing the ratio of allelic expression from the inactive X-chromosome (Xi) relative to the total allelic expression from both X-chromosomes (Xi + Xa) for **b** reactivated genes, for **c** full escape genes, and **d** late-silenced genes. Background colors represent the overall mRNA expression levels as Z-scores across different cell-types. M-ctrl iPSCs: $n = 6$, J-ctrl iPSCs: $n = 8$, NPCs and neurons of M-ctrl and J-ctrl: $n = 4$, A-ctrl iPSCs, NPCs and neurons: $n = 3$. The scatter plots show the mean allelic expression with the error bars indicating the 99% confidence interval. **e** Scheme showing mono-allelic and biallelic expression of target genes (pink spots) in the presence of XIST (yellow spot). **f** Single focal plane image showing RNA-FISH against *XIST* (yellow) and the pre-mRNA of *SPIN3* (magenta) in iPSCs (upper panel) and NPCs (lower panel) from the A-ctrl line. **g** Quantification of mono- and biallelic nuclear *SPIN3* signal in *XIST*-positive nuclei in the A-ctrl line. Biallelic *SPIN3* in iPSCs: 0.130; NPCs:

0.058. **h** Single focal plane image showing RNA-FISH against *XIST* (yellow) and the pre-mRNA of the full escape gene *NLGN4X* (magenta) in iPSCs (upper panel) and NPCs (lower panel) from the A-ctrl line. **i** Quantification of mono- and biallelic nuclear *NLGN4X* signal in *XIST*-positive nuclei in the A-ctrl line. Biallelic *NLGN4X* in iPSCs: 0.437; NPCs: 0.309. **j** Single focal plane image showing RNA-FISH against *XIST* (yellow) and the pre-mRNA of *MID1* (magenta) in iPSCs (upper panel) and NPCs (lower panel) from the A-ctrl line. **k** Quantification of mono- and biallelic nuclear *MID1* signal in *XIST*-positive nuclei in the A-ctrl line. Biallelic *MID1* nuclei in iPSCs: 0.114; NPCs: 0.320. For **f**, **h**, **j**, scale bar = 3 μm. For **g**, **i**, **k**, $n$ = individual cell nuclei. Statistical comparisons were facilitated with a Chi-squared test for association. All experiments were conducted using samples with at least 70% of *XIST*-positive cells. **l** Schematic illustration summarizing the observations of reactivation and late-silencing of X-linked genes during neural differentiation. Source data are provided as a Source Data file.

reactivated genes compared to inactive genes which were consistently monoallelic in our data (Fig. 2a). Recently, age-related reactivation of gene expression from the inactive X-chromosome has been described in the mouse[28]. As such ageing induced biallelic expression was concentrated at distal chromosomal regions, we investigated the distribution of the biallelic genes along the X-chromosome. Notably, reactivated genes did not accumulate at the distal ends of the X-chromosome. Yet, the biallelic genes were not randomly distributed but rather clustered together, indicated by significantly lower pairwise distances compared to a random background distribution (Fig. 2b, c). By contrast, such spatial organization along the X-chromosome was not observed for full-escapees and late-silenced genes.

Previous studies have reported distinct epigenetic signatures for genes that are subject to or escape XCI (reviewed in ref. 29). To gain insight into the epigenetic characteristics of the different categories of biallelic genes in our study, we therefore employed the *Roadmap Epigenomics 15-state chromatin model*[30]. Using this in-silico approach we analyzed the chromatin states at reactivated compared to inactive (monoallelic) genes in human female and male fetal brain tissue. This assessment revealed that reactivated genes were enriched for active transcription states ("Flanking active TSS", "Strong transcription" and "Weak transcription") (Fig. 2d) specifically in female brain tissue and not in male samples. An enrichment of the same epigenetic states was found for full escapees compared to inactive genes (Fig. 2d). While some states were also enriched in male fetal brain tissues, this effect was attributable to the pseudoautosomal-region (PAR) genes that have a homolog on the Y-chromosome. In line with their mono-allelic expression in NPCs and neurons, late-silenced genes were associated with repressed chromatin states in fetal brain tissue (Fig. 2d). To follow up on this analysis, we examined the pattern of key histone modifications in female fetal brain tissue obtained from ENCODE. We observed an increased signal for the active mark H3K36me3 over gene bodies for reactivated and full-escape genes with a concomitant decrease in the constitutive heterochromatin mark H3K9me3, which correlates with the enrichment of active chromatin states for these gene categories (Fig. 2e). Furthermore, late-silenced genes were associated with a higher H3K27me3 and H3K4me1 signal around their transcription start sites (TSS). These histone modifications are associated with bivalent/poised TSS. As late-silenced genes also showed decreased H3K9me3 signal compared to inactive genes, this pattern suggests that late-silenced genes are only facultatively silenced and not part of the constitutive heterochromatin as supported by their biallelic expression in undifferentiated iPSCs (Fig. 1d).

## Dynamic escape is conserved in the mouse
Among eutherian mammals, the mechanisms of X-linked dosage compensation vary[31]. However, recent evidence has drawn a picture of dynamic escape from X-inactivation also in the mouse[28,32,33]. Notably, the here described reactivated, full-escape and late-silenced genes were significantly enriched for genes previously reported to escape

X-inactivation in the mouse (Fig. 3a). In agreement with the diverse categories of biallelic genes that we observed in human differentiating cells, a recent study identified NPC-specific facultative escapees during mouse neural differentiation[32]. 50% of the genes reactivating during human neural differentiation in our study escape XCI either facultatively or NPC-specific in mice (Fig. 3b). Moreover, NPC-specific escapees in the mouse only overlapped with reactivated and late-silenced genes and not with full escapees. Together, these data suggest that differentiation-specific facultative escape is a broader feature of eutherian mammals rather than a trait specific to humans.

To follow up on this observation, we investigated allele-specific expression of the mouse homolog of the *MID1* gene, which we found to be reactivated during neural differentiation in human cells (Fig. 1j, k, Fig. S2c–e, Fig. S3i–l). By using RNAscope, we visualized the *Mid1* pre-mRNA in E13.5 mouse brain and skin tissues. It is important to mention that the *Mid1* locus has rearranged substantially between mouse and human. While in humans *MID1* is located on the short arm of the X-chromosome, distal to the PAR, in mice *Mid1* spans the PAR boundary (Fig. 3c) and the gene is considered to constitutively escape X-chromosomal inactivation[32]. Remarkably, single-molecule pre-mRNA FISH for the co-detection of the murine *Mid1* gene and *Xist* revealed that *Mid1* is monoallelically expressed in the majority of cells in both brain and skin, suggesting a more complex escape pattern at the single-cell level in the mouse embryo. Biallelic expression of *Mid1* was found in both tissues but significantly more cells exhibiting biallelic expression in the brain compared to skin (Fig. 3d, e), indicating that, similar to humans, *Mid1* is a facultative escape gene with a dynamic cell- and tissue-specific allele usage in mouse embryos. The increased biallelic expression in the developing mouse brain correlates with the differentiation-induced reactivation we observed in the human model and implies that inter-species conserved regulatory elements orchestrate tissue-specific escape from X-inactivation of *MID1*.

## Dynamic allele usage of X-linked genes in human developing tissue
To follow reactivation of gene expression from the inactive X-chromosome along neural differentiation on a single cell level, we generated brain organoids from two iPSC lines expressing high levels of *XIST* (hoik1, A-ctrl, see Supplementary Data 1, Fig. S4a) and performed single nucleus RNA-seq (snRNA-seq) from iPSCs and organoids (d30) (Fig. S4b). Force-directed graph embedding of the data allowed reconstructing differentiation trajectories from iPSCs via NPCs to multiple neuronal and glial fates (Fig. 4a, Fig. S4c–i). To follow the usage of the two X-chromosomes we pseudo-bulked the data and determined heterozygous X-chromosomal expressed SNPs for each cell line. We then constructed a variant-aware reference genome, to which we remapped the raw sequencing data to. We detected a total of 52 X-linked genes with a traceable SNP, i.e., an SNP located within 150 bp upstream of the polyA-tail. Despite the different cellular

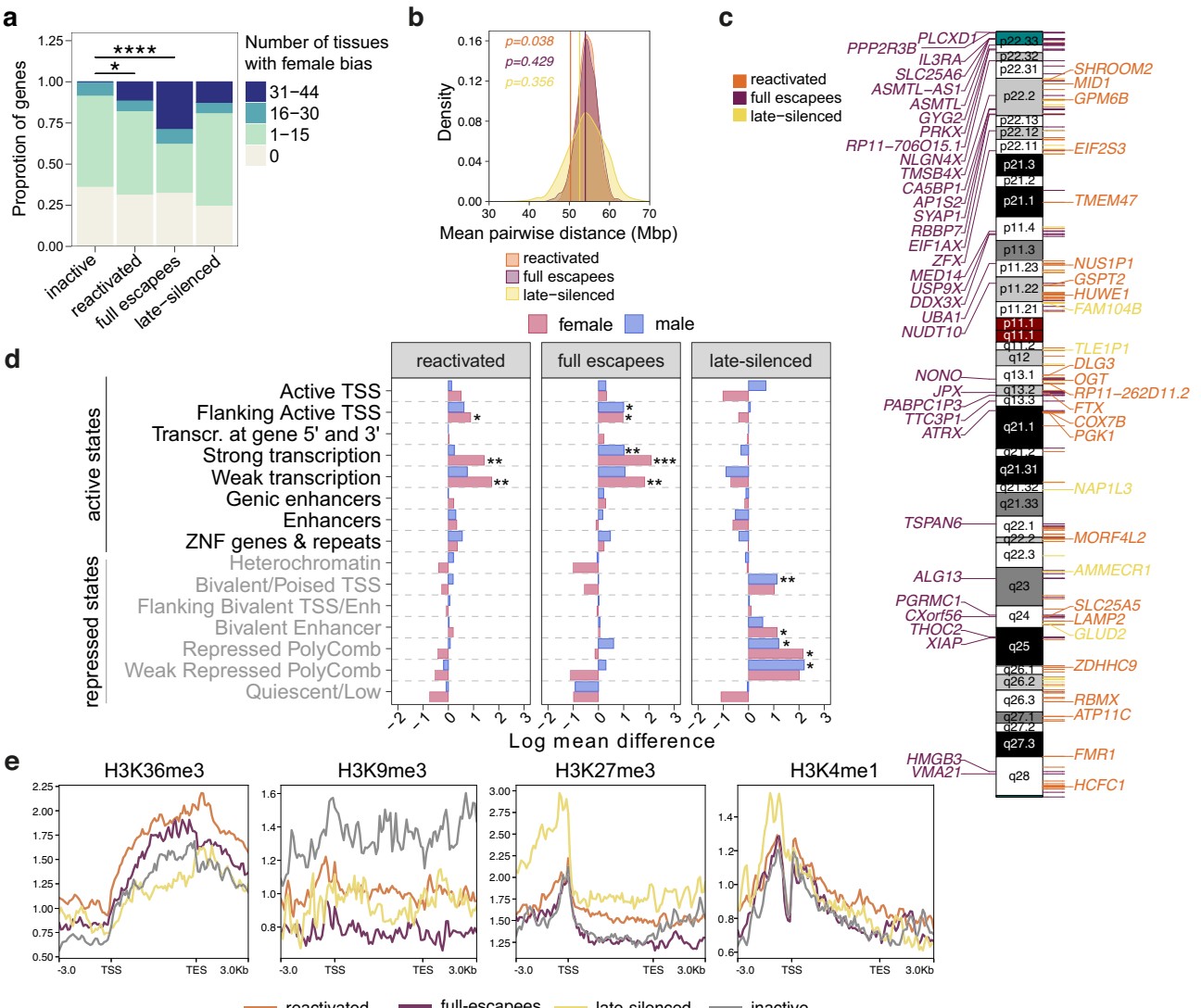

**Fig. 2 | Re-activating genes cluster and are associated with active chromatin.**
**a** Bar plot depicting the distribution of the proportion of genes with female sex bias in bins with different numbers of tissues for inactive, reactivated, full escape and late-silenced genes. Exact *P*-values (left to right): 0.0277, $6.32 \times 10^{-7}$, two-sided Fisher's exact test. **b** Density plot showing the background distribution of the mean pairwise distance between transcription start sites for 1000 sets of randomly selected X-linked genes. Due to varying sample sizes in the groups of reactivated, escape and late-silenced genes, a separate analysis was performed for each set of biallelically expressed genes. Vertical lines show the observed mean pairwise distance between reactivated, escape or late-silenced genes, respectively. Note that the reactivated but not full escape or late-silenced genes localize closer to each other than expected by chance. *P*-values were calculated using the cumulative distribution function of the normal distribution. Genes detected as biallelically expressed in at least one of the cell lines from the bulk RNA-seq analysis (A-ctr, J-ctrl or M-ctrl) were included in the analysis in (**a**, **b**). **c** Ideogram of the X-chromosome

showing the cytogenetic location of full escape, reactivated, and late-silenced genes across the three cell lines tested. Each colored line corresponds to a gene. Genes that were detected in at least two different cell lines are labeled with the gene symbol. The pseudoautosomal regions are depicted in turquoise on the ideogram. **d** Bar plots showing the log mean difference of chromatin state enrichment[30] between reactivated, escape and late-silenced compared to inactive (monoallelic) genes in female and male fetal brain tissue. Positive values indicate that the corresponding state is enriched in reactivated/escape/late-silenced genes compared to inactive genes. Exact *P*-values (top to bottom) reactivated: 0.01369, 0.00667, 0.00667; full-escapees: 0.01721, 0.01189, 0.00321, 0.00022, 0.00196; late-silenced: 0.00565, 0.01403, 0.04788, 0.01324, 0.02988; right-sided Wilcoxon rank sum test followed by Benjamini-Hochberg correction for multiple comparisons. **e** Metaplots showing histone modification signal over gene bodies for reactivated, full-escape, late-silenced and inactive genes in female fetal brain tissue samples obtained from ENCODE. Source data are provided as a Source Data file.

sources of the samples and the employed techniques, several of these genes overlapped with the genes detected in the bulk RNA-seq data (Supplementary Data 2). Biallelic expression was detected to be low in iPSCs but substantially increased along differentiation of *XIST* expressing lines to NPCs and in distinct neuronal lineages (Fig. 4b, c). Interestingly, when we aggregated the data in clusters using a PAGA representation (Fig. 4d), we found a highly variable degree of biallelic expression between the different clusters and trajectories, with an overall lower extent of dual allele usage at the onset of the

differentiation trajectories, i.e., the iPSCs (Fig. 4d). When plotting biallelic expression of X-chromosomal genes versus *XIST* levels we confirmed the previous RNA-FISH results detecting a considerable number of cells with high *XIST* levels that were actively transcribing genes from their inactive X-chromosome (Fig. 4e). To further address the relationship between biallelic expression and *XIST* expression, we analyzed the biallelic expression scores of the reactivating (monoallelic in iPSCs, biallelic in NPCs and more differentiated cells, Fig. 4f, left panel) and the full escape genes (biallelic in iPSCs and

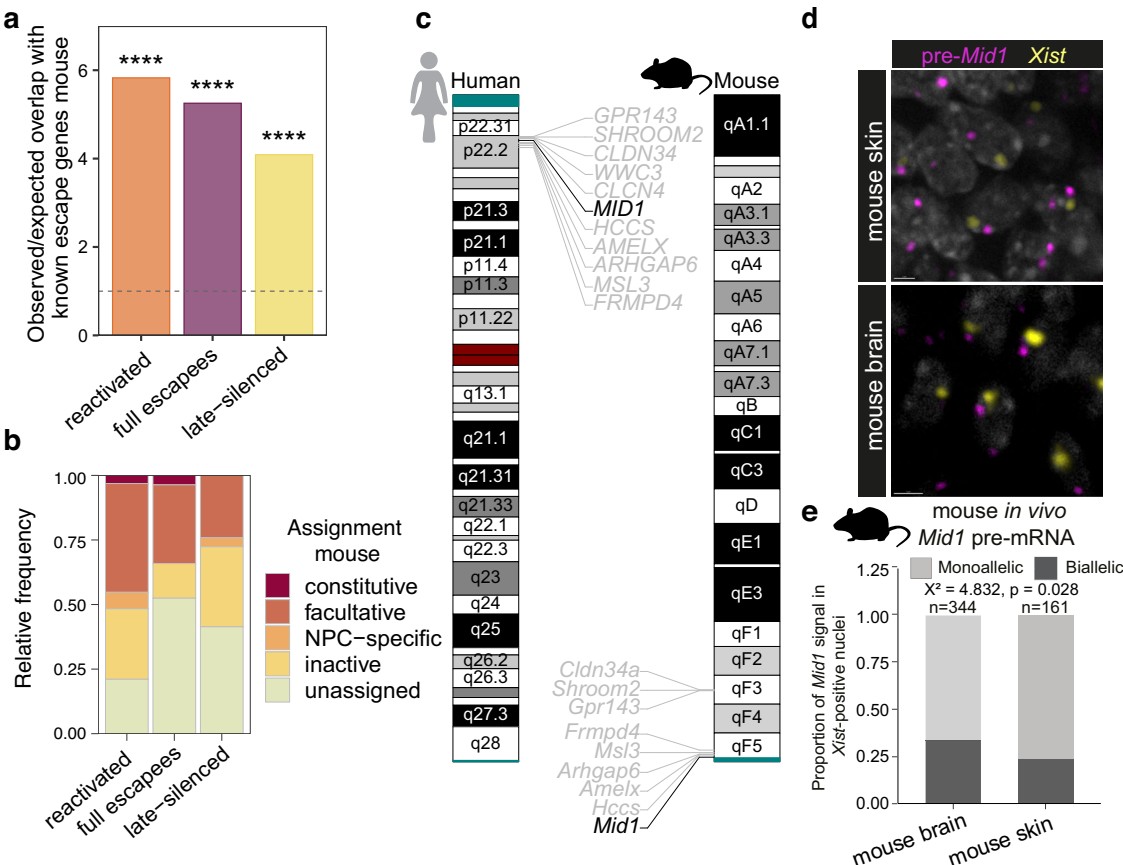

**Fig. 3 | Dynamic escape is conserved in the mouse. a** Bar plot showing the ratio of the observed to expected overlap of reactivated, full escape and late-silenced genes with previously reported escape genes in the mouse. Exact *P*-values (left to right): $2.2 \times 10^{-16}$, $2.2 \times 10^{-16}$, 5.736e-06. *P*-values were calculated using the cumulative distribution function of the normal distribution. **b** Bar plot showing the fraction of the assignment of reactivated, full escape and late-silenced genes in this study to different types of biallelic expression reported in the mouse[32]. **c** Ideograms depicting the cytogenetic location of *MID1* and the 10 surrounding protein coding genes in human compared to the mouse. The pseudoautosomal regions are

indicated in turquoise. **d** Single focal plane image showing RNA-FISH against *Xist* (yellow) and *Mid1* pre-mRNA in embryonic mouse brain (upper panel) and skin (lower panel) tissues at developmental stage E13.5. **e** Quantification of nuclear biallelic signal of *Mid1* in *Xist*-positive nuclei of embryonic brain and skin tissues. Fraction of biallelic *Mid1* nuclei in mouse brain: 0.337; skin: 0.236. *n* = individual cell nuclei from two different mice analyzed indicated above each barplot. Percentage of *Xist*-positive cells is 81% for brain tissue and 74% for skin tissue. Statistical analysis was performed by Chi-squared test for association. Source data are provided as a Source Data file.

differentiated cells, Fig. 4f, right panel) along differentiation pseudo-time in both *XIST* high expressing cells and those in which we could not detect *XIST* suggesting no or low (below detection threshold) *XIST* expression. While the data suggest that the absence of *XIST* impacts on the degree of biallelic expression of full escape genes, in particular later along the differentiation trajectories, we did not observe such an influence for reactivating genes. Furthermore, it became apparent that reactivation is most prominent at the progenitor level (marked by high *SOX2* and *NES* expression, Fig. 4f left panel). In sum, these data further corroborate the RNA-FISH results showing that biallelic expression induced upon neural differentiation is dynamic and lineage-specific across individual cells.

In a next step we took advantage of existing scRNA-seq data derived from the developing human spinal cord of two different embryonic stages to address allele usage of X-chromosomal genes in vivo during human CNS development. The Rayon dataset[34] includes transcriptomic data of single cells from the developing spinal cord of 6 female embryos across two developmental stages, Carnegie Stage (CS) 14 and CS19, i.e., several weeks after completion of XCI in humans. Following clustering and embedding of the data, distinct lineages of the CNS, the PNS, and paraxial mesoderm (Fig. 4g, Fig. S4j) were discernible. High *XIST* expression was observed across cells of all embryos

and an overall high but variable degree of biallelic expression of X-linked genes located outside of the pseudoautosomal region was detected (Fig. 4h, Fig. S4k). In order to delineate the extent of biallelic expression of genes in vivo, we grouped the cells according to their lineage and calculated the level of mean biallelic expression of the individual genes across different lineages (Fig. 4i, Supplementary Data 4). This analysis showed a striking lineage- and locus-specific dual allele usage. In line with the data derived from brain organoids, biallelic expression is highest at the progenitor stage and is diminishing along differentiation (Fig. 4j, Fig. S4l). Finally, these data also show an overall lower degree of biallelic expression in the cells from developmentally older embryos (CS19, Fig. 4k, Fig. S4m). Together with the observation that we again find biallelic expression in both, *XIST* high and *XIST* low expressing cells (Fig. 4k), this suggests that it is the differentiation stage rather than the expression of *XIST* that promote reactivation of gene expression from the inactive X-chromosome.

### X-reactivation impacts on phenotype dimorphism of OS
The observation that some reactivating genes cluster in NDD hotspot regions on the X-chromosome (Xq11.2[35,36], Fig. 2c), raised the question whether X-reactivation is a more general component associated with NDDs. Indeed, amongst the reactivating genes, but not full escapees

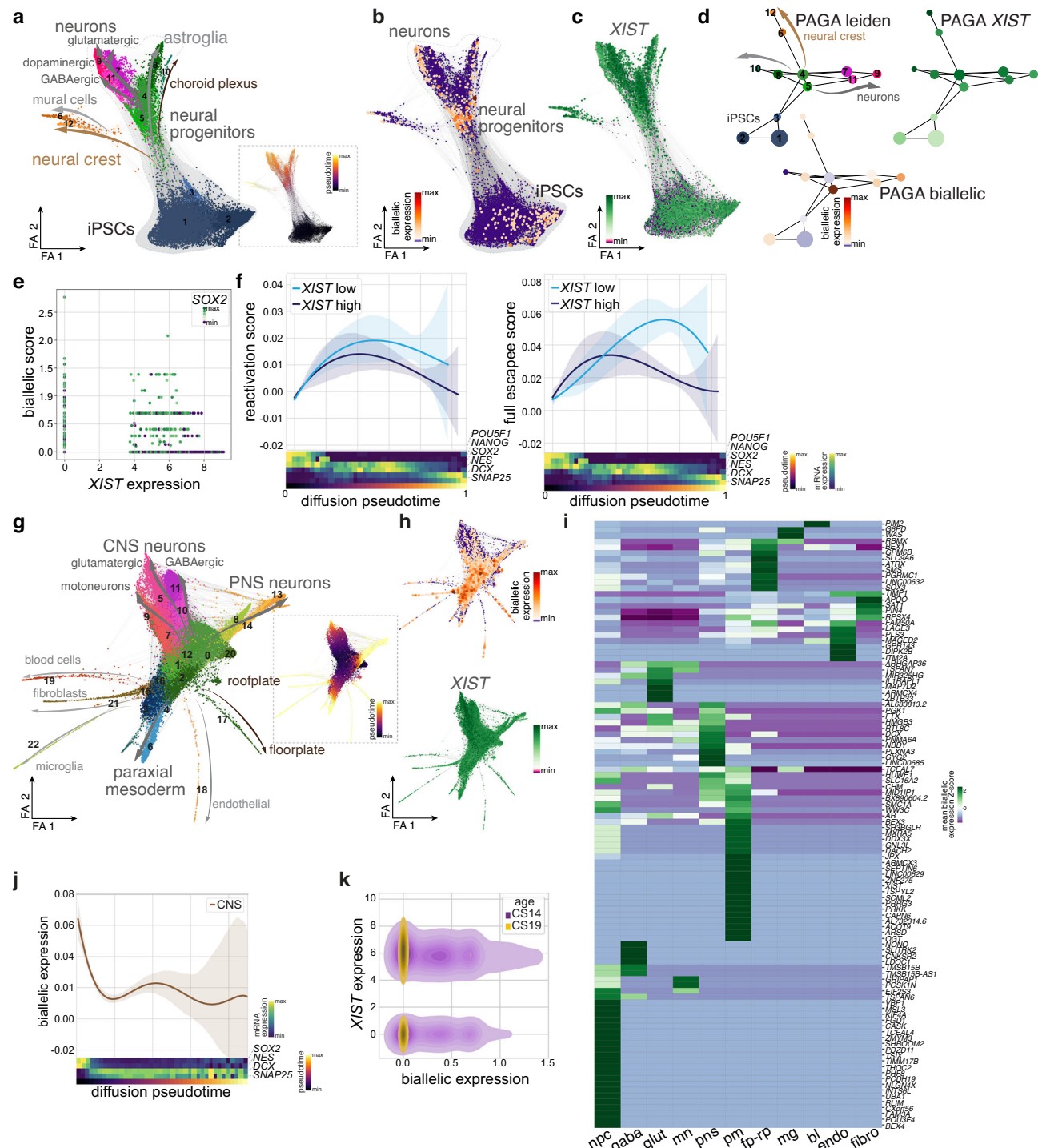

and late-silencing genes, we discovered an enrichment of NDD associated genes (Fig. 5a, Supplementary Data 2) beyond the expected overrepresentation on the X-chromosome[17]. Furthermore, disease enrichment analysis revealed that reactivating genes were associated with 'developmental disorder of mental health' and 'intellectual disability' (Fig. S5a). This suggests that in particular genes associated with neurodevelopment and NDDs were evolutionary endowed with dynamic X-reactivation as a regulatory mechanism. To investigate the molecular framework the reactivated genes are contributing to, we constructed a protein–protein interaction network based on reactivated genes expressed in NPCs and neurons (Fig. S5b). A gene ontology (GO) enrichment analysis revealed an overrepresentation of terms associated with RNA binding and epigenetic regulatory processes such

as histone binding as well as protein ubiquitination (Fig. S5c, Supplementary Data 3) highlighting the potential global impact on gene and protein function by the reactivation of X-linked genes. Remarkably, *MID1*, which we already validated as a reactivating gene during human neural differentiation and a facultative escapee in mouse embryos, was one of the hub proteins in the protein–protein interaction network.

To study the potential effect of reactivation of gene expression from the inactive X-chromosome on NDD-associated phenotypes, we employed fibroblasts from a female heterozygous carrier of a 4-bp deletion in the *MID1* gene, causative for Opitz BBB/G Syndrome (OS)[21–23], as well as fibroblasts from her affected male fetus. The fetus had inherited the 4-bp deletion and had prenatally developed OS. iPSCs of the female carrier and the male fetus (M-OS/male) were

**Fig. 4 | Dynamic allele usage of X-linked genes in human developing tissue.**
**a** Force-directed graph of snRNA-seq data from iPSCs and organoids (d30) using the A-ctrl and the hoik1 iPSC lines showing cluster and lineage annotations. Diffusion pseudotime is indicated in the dashed box. Number of cells total: 18.638, A-ctrl: 901, hoik1: 17.737. **b** Degree of biallelic expression of X-linked genes. **c** Expression of *XIST* in snRNA-seq data. **d** PAGA plot showing *XIST* expression and the degree of biallelic expression. **e** Dot plot showing in individual cells (all cells included in the embedding in Fig. 4a) the correlation between *XIST* expression and biallelic score. The color code indicates the expression level of the progenitor marker *SOX2*. **f** Fitted line plots indicating the reactivation score (left) and the full escapee score (right) in *XIST* positive cells along diffusion pseudotime. 95% confidence interval is shown as a transparent band in the respective same color. On the X-axis the expression levels of selected marker genes are shown along pseudotime. **g** Force-directed graph embedding of published scRNA-seq data[34] showing the clusters of six female human developing spinal cord samples. Number of cells total: 39.210.

Inset shows diffusion pseudotime. **h** Degree of biallelic expression (upper plot) and *XIST* mRNA level (lower plot). In a, b, c, g, h FA: force atlas. **i** Heatmap depicting the Z-score of biallelic expression of all non-pseudoautosomal genes detected to be biallelically expressed in distinct differentiation lineages. npc (neural progenitor cells), gaba (GABAergic neurons), glut (glutamatergic neurons), mn (motoneurons), pm (paraxial mesoderm), pns (peripheral nervous system), fp-rp (floorplate-roofplate), mg (microglia), bl (blood cells), endo (endothelial cells), fibro (fibroblasts), summarized in Supplementary Data 4. **j** Line plot showing biallelic expression of non-pseudoautosomal genes along pseudotime in the CNS lineage (central nervous system, clusters 0, 1, 5, 7, 9, 10, 11, 12). 95% confidence interval is shown as a transparent band in the respective same color. On the X-axis the expression levels of selected marker genes are shown along pseudotime. **k** Density plot showing biallelic expression and *XIST* expression in the same cells separated by developmental age. CS Carnegie Stage.

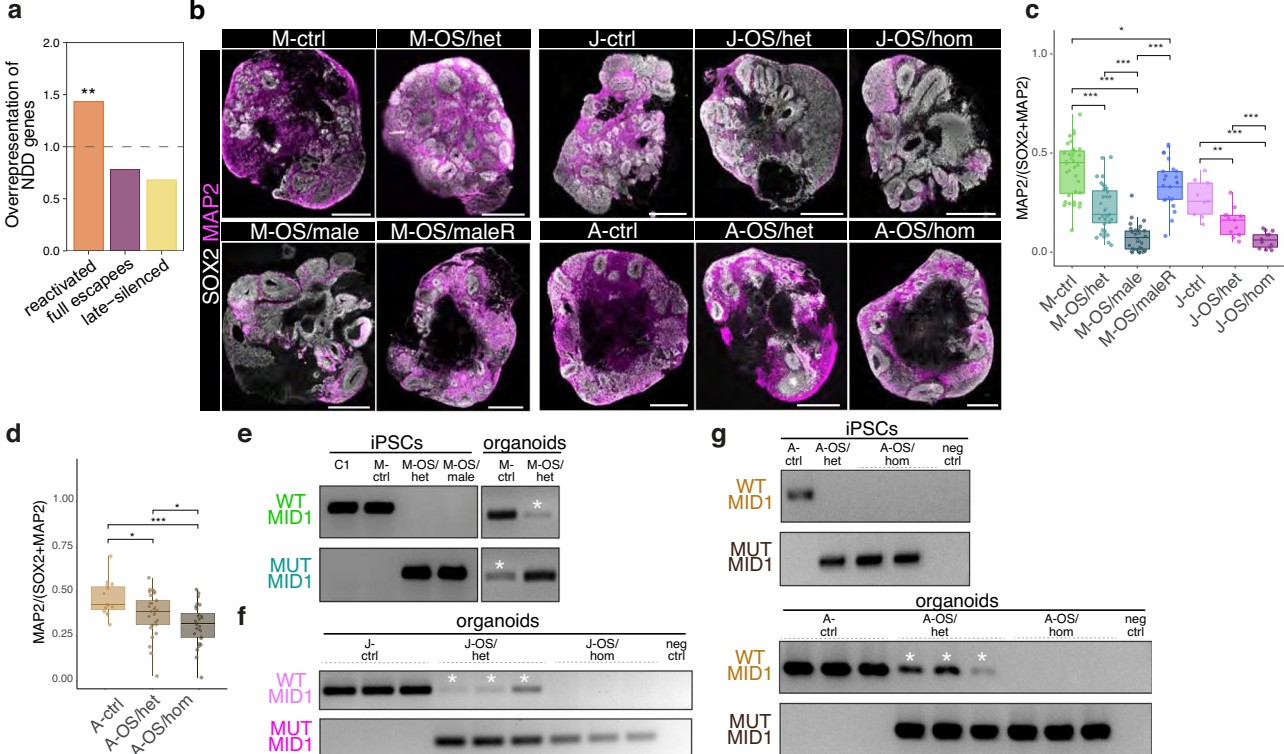

**Fig. 5 | X-reactivation impacts on phenotype dimorphism of OS. a** Bar plot showing the ratio for the observed relative to the expected overlap of reactivated, full escape and late-silenced genes with high-confidence NDD genes. For reactivated genes, an intriguing accumulation of NDD genes was observed (1.43 times more than expected; $P = 6.72 \times 10^{-3}$). *P*-values were determined by the cumulative distribution function. **b** Immunostaining of organoid sections of the different experimental lines showing SOX2 (white) and MAP2 (magenta). Scale bars = 500 μm. **c** Box and jitter plots showing the quantification of the fraction of MAP2 in neural tissue (SOX2 + MAP2+ area) on d30 (M-ctrl, $n = 36$; M-OS/het, $n = 39$; M-OS/male, $n = 34$; M-OS/maleR, $n = 25$; J-ctrl, $n = 9$; J-OS/het, $n = 13$; J-OS/hom, $n = 13$; exact *P* values (top to bottom): 0.03, $9.7 \times 10^{-13}$, $2.4 \times 10^{-9}$, $<2.2 \times 10^{-16}$, $8.1 \times 10^{-9}$, $7.9 \times 10^{-4}$, $4.0 \times 10^{-6}$, $4.3 \times 10^{-3}$). **d** Box and jitter showing the quantification of

the fraction of MAP2 in neural tissue (SOX2 + MAP2+ area) on d30 in the *XIST* expressing A-lines (A-ctrl, $n = 15$; A-OS/het, $n = 29$; A-OS/hom, $n = 28$; exact *P* values (top to bottom): 0.035, $8 \times 10^{-5}$, 0.032). For **c**, **d** boxplots show median, quartiles (box), and range (whiskers). In the jitter plots dots represent individual organoids. Two-sided Wilcoxon rank sum test; *$*P < 0.05$, **$P < 0.01$, ***$P < 0.001$, ns $P > 0.05$. **e** Allele-specific RT-PCR revealing reactivation (*) of the inactive *MID1* allele in brain organoids derived from the M-lines. **f** Reactivation (*) of the inactive *MID1* allele in brain organoids derived from the J-lines as revealed by allele-specific RT-PCR. **g** Reactivation (*) of the inactive *MID1* allele in brain organoids derived from the A-lines as revealed by allele-specific RT-PCR. Source data are provided as a Source Data file.

generated. The clonal nature of human iPSCs maintaining the X-inactivation status of the cell of origin[14] in conjunction with the possibility to trace the expression of the 4-bp deletion enabled us to establish patient-derived female iPSCs carrying the 4-bp deletion variant either on the inactive X-chromosome (M-ctrl) or on the active X-chromosome (M-OS/het). Moreover, we employed CRISPR/Cas9 to additionally engineer independent female iPSC lines (J-ctrl, A-ctrl) introducing the 4-bp deletion on the active X-chromosome (J-OS/het,

A-OS/het) or on both X-chromosomes (J-OS/hom, A-OS/hom) (Fig. S6). We further included the male patient lines before (M-OS/male) and after CRISPR/Cas9-mediated repair (M-OS/maleR) in our experimental setup (Fig. S5d–i, Supplementary Data 1). To analyze if X-chromosomal reactivation of the inactive wildtype allele has an influence on the phenotypic presentation of X-linked OS, we generated brain organoids from these iPSC lines carrying OS-causing *MID1* mutations (Fig. 5b). Inspection of the cellular organization of the brain organoids at day 30

(d30) revealed an increase in the area covered by NPC-containing ventricular zone-like structures (VZLS) in *MID1* mutant organoids (Fig. S7a, b) while the cell density remained unchanged (Fig. S7c). Quantification of the cellular composition of brain organoids showed a decrease in MAP2$^+$ neurons within *MID1* mutant organoids (M-OS/het, M-OS/male, J-OS/het, J-OS/hom, A-OS/het, A-OS/hom) compared to *MID1* wildtype organoids (M-ctrl, J-ctrl, A-ctrl) (d30: Fig. 5b-d, Fig. S7d; d50: Fig. S7e–g), recapitulating microcephaly and hypoplasia of the cerebellum observed in OS patients. Importantly, CRISPR/Cas9-mediated repair of the *MID1* mutation in the male cells (M-OS/maleR) led to a rescue of the phenotype (Fig. 5b, c, Fig. S7a, b, d) strongly supporting MID1 dependency of the phenotypic effects.

To directly assess whether altered cell cycle dynamics contribute to the development of the OS phenotype, BrdU pulse-chase experiments were conducted in d30 brain organoids (Fig. S7h). We determined the number of postmitotic cells (Ki67$^-$) that had incorporated the thymidine analog BrdU (BrdU$^+$, i.e., passed through S-phase) during the time of the pulse-chase experiment. Quantification of BrdU$^+$ Ki67$^-$ cells within the organoid VZLS showed a reduced cell cycle exit of NPCs in OS organoids and a concomitant decrease of cells that differentiated into neurons (BrdU$^+$ NeuN$^+$) (Fig. S7h–m). In addition, more BrdU$^+$ cells were retained within the VZLS (Fig. S7n) correlating with the observed increase of the VZLS area in OS organoids (Fig. S7a, b). These data indicate that MID1 controls the balance between proliferation and differentiation of NPCs.

Strikingly, the brain organoids derived from the heterozygous female iPSC lines, M-OS/het, J-OS/het and A-OS/het, carrying the mutant *MID1* allele on the active X-chromosome, exhibited a less severe differentiation phenotype compared to M-OS/male (Fig. 5b-d, Fig. S7a, b, d). While this is reminiscent of the clinical manifestations of X-linked diseases, in our paradigm using clonal iPSC lines with a defined and stable X-(in)activation status random X-inactivation cannot contribute to the amelioration of the phenotype in female tissue. To assess if the difference in the phenotype severity correlates with the reactivation of the wildtype allele on the inactive X-chromosome, we used allele-specific RT-PCR of the *MID1* locus. Substantial reactivation of the *MID1* allele on the inactive X-chromosome was found in NPCs, neurons (Figs. S2c-e, S8a, b) and brain organoids (Fig. 5e-g) derived from all heterozygous cell lines. The presence of H3K27me3 territories and *XIST* expression also upon CRISPR/Cas9-mediated genome editing was confirmed in the A-ctrl, A-OS/het and A-OS/hom lines (Figs. S3b, S8c). Allele-specific RT-PCR analysis of *MID1* showed that it kept monoallelic expression in iPSCs up to high passages in low *XIST* expressing iPSC lines (Fig. S8d), indicating that it is not a target of erosion. Western blot analysis confirmed that reactivation of the wildtype allele from the inactive X-chromosome resulted in the expression of MID1 wildtype protein in the OS/het NPCs, while no wildtype MID1 protein expression was detected in the corresponding iPSCs (Fig. S8e, f). To test the impact of the reactivation of the wildtype *MID1* allele on the inactive X-chromosome in heterozygous female OS cells on the phenotype manifestation, we took advantage of the female iPSCs homozygous for the 4-bp deletion (J-OS/hom, A-OS/hom) (Fig. S5d). Brain organoids derived from these lines showed a more severe phenotype than the OS/het lines (J-OS/het, A-OS/het), comparable to the male OS organoids (Fig. 5b-d, Fig. S7a, b, d). Interestingly, the presence of a wildtype allele on the inactive X-chromosome ameliorates the disease phenotypes in *XIST* high (A-line) and low (M- and J-lines) expressing cells. These data demonstrate that the presence of a wildtype allele on the inactive X-chromosome in heterozygous mutant female cells, curtails the severity of the disease phenotype through reactivation of the allele on the inactive X-chromosome.

To identify the molecular signatures underlying the gradual OS phenotypes, we performed scRNA-seq of cells derived from brain organoids (d30) generated from the M- (M-ctrl, M-OS/het) and the J-lines (J-ctrl, J-OS/het, J-OS/hom) and focused our attention on the

neural lineages (Fig. S9a, b). Within the neural tissue we found cells with molecular signatures of neural crest, choroid plexus, midbrain, and hindbrain lineages (Fig. S9b, c). This hindbrain organoid model recapitulated the major lineages of the developing cerebellum, formation of which is disturbed in OS patients as well as in the mouse model[37,38]. NPC-to-neuron trajectories producing distinct neurotransmitter identities were discriminated: rhombic lip progenitors (*ATOH1*) giving rise to glutamatergic cells expressing *SLC17A6*, VZ progenitors (*PTF1A*) differentiating into GABAergic cells expressing *GAD2* (Fig. S9c). To elucidate the molecular framework driving the early disruption of the balance between proliferation and differentiation observed in OS brain organoids (Fig. 5b-d), differentially expressed genes in NPCs (clusters 0, 1, 2, 3, 4, 5, Fig. S9d) between ctrl and OS lines were determined followed by GO analysis (Fig. S9e, f). Amongst the differentially expressed genes, we found *ZIC1* and *ZIC2* (Fig. S9d). Mutations in *ZIC1* and *ZIC2* lead to Dandy-Walker cysts[39] which are characteristic malformations of the cerebellum. Interestingly, similar to *MID1,* these genes regulate neuronal differentiation[40]. Hence, our data suggest a partial overlap and a surprising molecular crosstalk linking OS and Dandy-Walker pathogenesis. Moreover, compared to wildtype, OS NPCs showed *MID1* genotype dependent gradual deregulation of more general cell cycle regulators associated with proliferation and differentiation (e.g., *CCND1, NIN, CDK6, KIF2A, CDKN1A, CDC42,* Fig. S9g)[41–45], providing a potential molecular framework underlying the altered cell cycle progression observed in OS organoids (Fig. 5b-d, Fig. S7b, d).

The severity of the phenotype in OS brain organoids (Fig. 5) varied depending on the presence (M-OS/het, J-OS/het, A-OS/het) or absence (J-OS/hom, A-OS/hom) of a wildtype *MID1* allele on the inactive X-chromosome with heterozygous lines showing intermediate phenotypes. To disentangle these tissue level phenotypes with cellular resolution we further leveraged the scRNA-seq data and computed the transcriptional deviation of each cell from their respective controls. These analyses revealed that the transcriptomes of cells in the cerebellar clusters (0, 2, 3, 4, 5, 6, 8, 10) differed strongest while the transcriptomes of midbrain (14), choroid plexus (15, 16) and neural crest (17, 18, 19) derived cells varied less upon *MID1* mutation (Fig. S9h) with slight variations between M- and J-lines. Importantly, lines with a wildtype *MID1* allele on the inactive X-chromosome (J-OS/het) exhibited an intermediate transcriptional phenotype reminiscent of the characterization of tissue composition (Fig. 5b-d, Fig. S7a, b, d). To temporally resolve the impact of the *MID1* mutation on the NPC-to-neuron differentiation trajectory of hindbrain lineages we plotted the degree of transcriptional deviation along the developmental pseudotime (Fig. S9i). The deviation from controls of both, heterozygous and homozygous lines was strongest early in the lineage at the NPC stage with high expression of the NPC marker *SOX2* and low expression of the mature neuronal marker *SNAP25* (Fig. S9i). These transcriptional differences occurred concomitant with an accumulation of cells at a transcriptional NPC stage in OS lines (Fig. S9j) in accordance with the increase of VZLS and SOX2 in OS organoids (Fig. S7a, b, d). The observation that cells with homozygous *MID1* mutations (J-OS/hom) react stronger to the lack of functional *MID1* and heterozygous lines (M-OS/het, J-OS/het) show intermediate transcriptional phenotypes reinforces the gradual manifestation of the phenotype on an individual cell level.

## The MID1 target PAX6 correlates with OS phenotype attenuation

With the goal to identify a putative molecular target of the ubiquitin ligase MID1 underlying the reduced neuronal differentiation and progenitor cell cycle progression phenotype, we set out to further analyze the progenitor cell population within the VZLS of the J- (low *XIST* expression) and the A-line (high *XIST* expression) organoids with cellular resolution. It has been shown previously that PAX6 can be

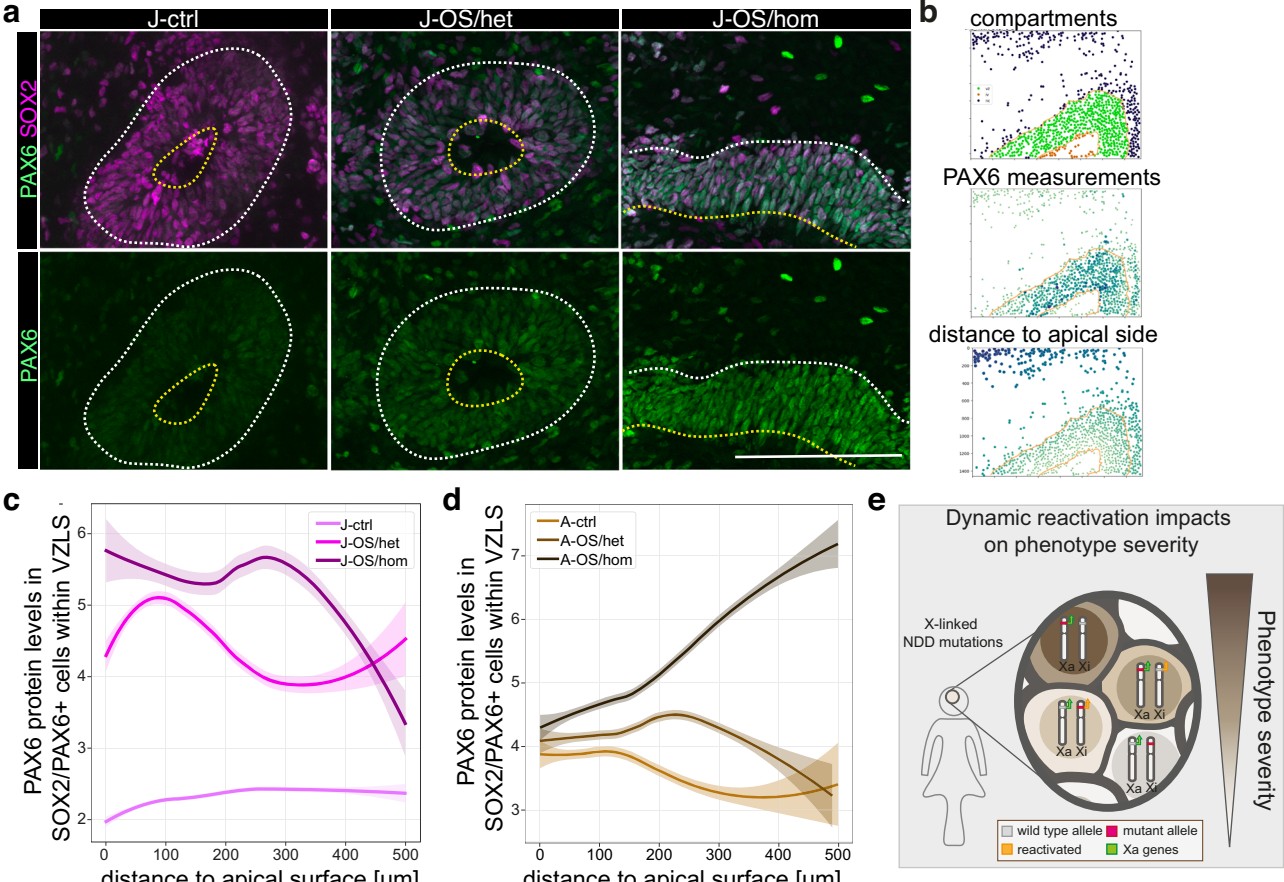

**Fig. 6 | The MID1 target PAX6 correlates with OS phenotype attenuation.**
**a** Immunofluorescence stainings using antibodies against PAX6 (green) and SOX2 (magenta) across experimental lines exemplifying the increased expression of PAX6 in VZLS (marked by dashed lines) in J-OS/het and more pronounced in J-OS/hom brain organoids. The apical site of the VZLS is indicated through a yellow dashed line. Scale bars = 100 μm. **b** Example of the readout of the python-based image analysis showing segmentation of nuclei within and outside of VZLS. **c** Line plots showing the levels of PAX6 protein in PAX6/SOX2 double positive VZLS cells in relation to the distance to the apical surface. Note the gradual increase in PAX6 protein levels dependent on the presence (J-OS/het) or absence (J-OS/hom) of the *MID1* wildtype allele on the inactive X-chromosome. $n = 5$ organoids in all conditions. J-ctrl=5074 cells, J/OS-het=5745 cells, J/OS-hom=5998 cells. **d** Line plots

showing the levels of PAX6 protein in PAX6/SOX2 double positive VZLS resident cells in relation to the distance to the apical surface in the *XIST* expressing A-lines. Note the gradual increase in PAX6 protein levels dependent on the presence (A-OS/het) or absence (A-OS/hom) of the *MID1* wildtype allele on the inactive X-chromosome. A-ctrl: $n = 3$ organoids; A/OS-het, A/OS-hom: $n = 4$ organoids. A-ctrl=4719 cells, A/OS-het=8506 cells, A/OS-hom=10408 cells. For **c**, **d** the plots show a nonparametric locally weighted regression (LOESS) fit as a solid line for each condition with 95% confidence interval shown as a transparent band in the respective same color. **e** Schematic highlighting that resilience of female tissues is enhanced through the possibility to reactivate genes on the second X-chromosome, thereby ameliorating the phenotype of X-linked NDDs. Green = mutant, pink = wildtype, light blue = reactivated. Source data are provided as a Source Data file.

targeted by MID1 for proteasomal degradation[46]. We hypothesized that also in human neural cells, the NPC regulatory protein PAX6 would be affected by the absence of MID1. In fact, we found a marked increase in the PAX6 protein levels within the VZLS of both *XIST* low J-OS/het and *XIST* high A-OS/het organoids, which was further increased in J-OS/hom and A-OS/hom organoids (Fig. 6a–d). Modulation of the cell cycle regulator and PAX6 target p21 is a possibility by which PAX6 can control the proliferation rate of NPCs[47]. We found p21 protein levels increased in the VZLS resident progenitors (Fig. S10a–c), and with similar correlation between PAX6 and the p21 protein levels in all three genotypes (Fig. S10d), resulting in an increase of non-apical mitotic NPCs (Fig. S10e–g). Together these data underscore the notion that the presence and reactivation of a wildtype allele on the inactive X-chromosome ameliorates the disease phenotypes not only on a cellular but also on the molecular level.

In sum, our data show that the reactivation of the expression of alleles from the inactive X-chromosome leads to a dynamic increase in the diversity of the available allele pool. Furthermore, our data demonstrate that in X-linked NDDs, this mechanism can enhance resilience of female neural tissue beyond the general random X-inactivation in tissues by ameliorating the phenotype on an individual cell level (Fig. 6e).

## Discussion

Escape from X-inactivation resulting in biallelic expression and often in a female expression bias, is a prime candidate molecular mechanism for sexual dimorphism in disease development[8,13,48]. In humans, however, studies addressing the developmental dynamics of X-chromosomal gene inactivation have been impeded by the difficult access to primary material and often restricted to readily accessible cells such as fibroblasts and lymphocytes or postmortem material. Consequently, the resulting analytical snapshots only represent single time points disregarding developmental trajectories. In an experimental set-up combining cell reprogramming and single human iPSC clone selection with in vitro neural differentiation protocols we were able to follow mono- and biallelic expression of X-chromosomal genes across differentiation. We found that reactivation and late-silencing of gene expression from the inactive X-chromosome leads to a highly

dynamic locus- and lineage-specific usage of X-chromosomal alleles induced by neural differentiation. This dynamic usage of alleles from the inactive X-chromosome was corroborated by snRNA-seq data of brain organoids and further confirmed in a re-analysis of publicly available scRNA-seq in vivo data of the developing human spinal cord.

While brain organoids are a more elaborate system providing a tissue-like context to model early developmental process, they are composed of a plethora of different cell types (neural stem cells, neurons, astrocytes, choroid plexus cells, neural crest cells, melanocytes). The 2D differentiation to NPCs and neurons provides a much more homogenous system in which the vast majority of the cells are either NPCs or neurons. The huge technical advantage of such a homogenous system is their amenability to bulk approaches to measure biallelic expression, such as bulk RNA-seq and QUASEP. In particular the bulk RNA-seq allows to describe the transcriptome of a system much deeper which is particularly important for the detection of expression from the inactive X-chromosome, because those transcripts are significantly less abundant than the ones from the active X-chromosome. A further big advantage of the 2D differentiation approach is that we can use pre-mRNA FISH to detect biallelic expression and corelate it with *XIST* expression on a single cell level. Importantly, pre-mRNA FISH is a completely independent method to validate the findings from bulk RNA-seq and single cell RNA-seq in organoids and human in vivo tissue. Together, our data strongly suggest that constitutive and facultative escape are distinct entities and provide evidence that reactivation and late-silencing lead to facultative escape in a tissue-, lineage- and developmental time-specific manner.

The spatial organization of full escape, reactivating and late-silencing genes along the X-chromosome, that we have identified in this study, differs. Full escapees were found to cluster particularly on the short arm of the X-chromosome[4,19,29,49–51]. This is related to X-chromosome evolution and the assumptions that the short arm of the X-chromosome has developed from an autosomal region distinct from the long arm[19] and that it is influenced by the pseudoautosomal region 1 (PAR1), that is homologous to Y-chromosomal sequences and as a whole constitutively escapes X-inactivation[29,52]. While we detected a higher number of full escapees on the short arm of the X-chromosome, the linear proximity between these genes was not different than expected by chance. This might be due to a bias in the analysis that depends on expressed SNPs in the used cell systems, which makes some genes inaccessible for the analysis. However, the pairwise distance between reactivating genes was lower than expected by chance suggesting clustering rather than random distribution of these genes (Fig. 2b, c). We found clusters of reactivating genes across the entire chromosome on both arms. This seems different to genes that are reactivating expression from the inactive X-chromosome during the aging process and that cluster predominantly at the chromosome ends[28]. In addition to the chromosomal localization, we assessed the epigenetic landscape of the reactivating genes compared to genes which had not been detected to show biallelic expression along the neural differentiation trajectory. Strikingly, we found reactivating genes in contrast to inactive genes to be associated with an epigenetic pattern of active transcription in female but not in male fetal brain tissue suggesting distinct chromatin states in the reactivation process. In contrast, late-silenced genes were enriched for facultative but not constitutive heterochromatin states and histone marks.

The spatial proximity of the reactivated genes together with their epigenetic signatures in vivo, suggest that facultative escape is achieved by a dynamically changing chromatin status, allowing locus-, lineage- and developmental time-dependent reactivation and late-silencing of X-chromosomal genes induced by differentiation.

Despite fundamental differences in X-inactivation mechanisms between mouse and human[2,10–12], stimulus-induced reactivation of gene expression from the inactive X-chromosome appears to be a phenomenon that is conserved across species. This notion is supported by our own RNA-FISH data on mouse embryos and recent studies describing development specific biallelic expression of X-chromosomal genes[32] together with a dynamic reactivation of selected genes during the aging process in the mouse[28,33].

*XIST* is the major player during X-inactivation. During prolonged culturing of ES cells and iPSCs *XIST* loss leads to erosion of inactivation across large parts of the X-chromosome[25–27,53–59] and, as shown recently, ectopic expression of *XIST* can reverse escape from X-inactivation in mouse NPCs[32]. In our experiments we have used *XIST* high and low expressing iPSCs, NPCs, neurons, brain organoids, as well as human embryos and detected reactivation and late-silencing of gene expression from the inactive X-chromosome at comparable levels in *XIST* high and low cells. Finally, facultative escape of the *Mid1* gene was confirmed in (*Xist* high expressing) E13.5 embryonic mouse skin and brain cells (Fig. 3d, e). Together, these data provide evidence that reactivation and late-silencing of X-chromosomal genes, other than full escape, rather depend on the developmental time point than on *XIST* expression levels. In line with our observation of a stimulus driven influence on gene expression from the inactive X-chromosome, a significant increase of expression from the inactive X-chromosome in the aging mouse was described[28,33]. Such a female biased expression of genes involved in synaptic transmission and memory formation can contribute to a higher cognitive resilience of the aging female brain[33]. Our results add the developmental perspective to these findings suggesting that dynamic and selective allele usage of X-chromosomal genes is a mechanism of fine-tuned gene expression regulation early on and throughout lifetime.

Interestingly, within the group of reactivating genes but not full escapees and late-silencing genes we found a surprising over-representation of NDD associated genes, linking reactivating genes to the development of NDD phenotypes. NDDs present with a substantial gender bias with females being less frequently and less severely affected than males[15]. Random X-inactivation and escape have been suggested to play a role in phenotype development of NDDs[17]. Our findings show that particularly NDD-associated genes were equipped with the ability to dynamically switch between mono- and biallelic expression. To test the functional implication of the reactivation of X-linked genes on NDD phenotypes, we used brain organoids to model the disease etiology of OS, an X-linked NDD caused by mutations in the *MID1* gene. We compared brain organoids derived from female iPSC clones that exclusively expressed mutant *MID1* (the wildtype allele was silenced on the inactive X-chromosome) with male cells carrying the same loss-of-function mutation in *MID1*. We found that the derived female neural tissue showed milder phenotypes than mutant male tissue suggesting a protective mechanism in female cells. We could further show that indeed reactivation of the wildtype *MID1* allele from the inactive X-chromosome during neural differentiation underlies the attenuation of the phenotype in female, heterozygous tissue. This is similar to what has been achieved artificially by epigenome editing of the wildtype *MECP2* allele on the inactive X-chromosome in Rett syndrome human neurons[60]. Our data propose that reactivation of the second allele from the inactive X-chromosome is a potent mechanism for female cells to curtail NDD phenotype development and thereby likely increase brain resilience. However, the reactivation of mutant alleles from the inactive X-chromosome might also have the potential to negatively influence disease development. Severe X-linked *HUWE1*-associated encephalopathy for example has been found to develop despite preferential inactivation of the mutated X-chromosome[61]. Reactivation of alleles carrying gain-of-function *HUWE1* mutations from the inactive X-chromosome in the brain could be the riddle's solution here.

Our data furthermore imply that X-linked gene reactivation not only influences X-chromosomal NDD phenotypes, but through their gene- and protein interaction partners the reactivating genes impact on a protein network with more global implications for the cells. In a

GO enrichment analysis of proteins interacting with the reactivating genes we found an overrepresentation of terms involved in two central control mechanisms: gene expression control on an epigenetic level on one hand and protein function regulation through the ubiquitin system on the other. Both systems are sensitive to dose changes[62–65]. Changes in expression levels through reactivation of the silent allele on the inactive X-chromosome would thus interfere with the interaction network. It is therefore conceivable that through a complex symphony of interactions conducted by the X-reactivation network the epigenetic machinery in female cells allows to modulate the expression not only from gonosomes but also autosomes. NDDs are frequently caused by single dominant heterozygous mutations or by the sum of several common variants with smaller disease effects in autosomal genes[66]. In mouse models, we and others have shown that through exogenous epigenetic modulation expression from the wildtype allele in diseases caused by heterozygous autosomal mutations can rescue disease phenotypes[67,68]. This raises the possibility that the X-reactivation network installs a safety net in the developing female brain increasing resilience towards NDDs and could be part of a mechanistic explanation for higher mutational burden in females required for the clinical manifestation of NDDs[18].

In the future, it will be interesting to see if signals other than differentiation and aging can stimulate X-chromosomal reactivation. Depending on the quality and quantity of such stimuli it is conceivable that reactivation may vary individually. It is attractive to hypothesize that individually differential reactivation has a graded influence on NDD phenotype development and might therefore be a fundamental reason underlying clinical variability of NDDs.

Taken together, our data describe a hitherto undiscovered mechanism of differentiation-dependent gene regulation in female brain cells leading to a dynamic increase in the diversity of the available allele pool and having the potential to modulate NDD phenotypes.

## Methods

### Human fibroblast cell lines
Skin punch biopsies (4 mm) were taken in the Children's Hospital of the University Medical Center in Mainz. Primary human fibroblasts were isolated from skin punch biopsies as previously described with small adaptations[69]. Briefly, biopsies were cut in small pieces containing all skin layers and plated on a 6-well plate coated with 0.1% gelatin with fibroblast extraction media (FEM; DMEM (Thermo Fisher), 20% fetal bovine serum (FBS, Thermo Fisher), 1% penicillin/streptomycin (P/S) (Thermo Fisher)). Cells were monitored daily, and fresh medium was added every other day. Fibroblasts started to migrate out of the skin biopsies after 7–10 days and were transferred to two T75 flasks after 3–4 weeks using TrypLE™ Express enzyme (Thermo Fisher). Fibroblasts were then cultured in IMDM (Thermo Fisher), 15% FBS, 1% P/S. When reaching 90% confluency, fibroblasts were replated into T175 flasks and expanded as needed or frozen in liquid nitrogen for long-term storage. M-lines: Patients' fibroblasts (for M-lines) had been established at Bristol Genetics Laboratory following the same procedure as described above.

### Reprogramming to pluripotent stem cells
Fibroblasts were reprogrammed into iPSCs with retroviruses and by using the feeder-dependent approach of the CytoTune(TM)-iPS 2.0 Sendai Reprogramming Kit (Thermo Fisher) according to the manufacturer's protocol. HEK 293 T cells used for retrovirus production were cultured in IMDM, 10% FBS and 1% Pen/Strep and replated every three to four days at a ratio of 1:10 or 1:20 by using TrypLE™ Express enzyme. To generate the retroviruses for reprogramming, HEK 293 T cells were transfected by using Lipofectamine 3000 (Thermo Fisher) following the manufacturer's instruction. Fibroblasts were transduced by retroviral spinfection. Briefly, 24 h prior to transduction

$1 \times 10^5$ cells fibroblasts were seeded on a 6-well plate coated with 0.1% gelatin and cultured in IMDM plus 15% FBS. On day 0, medium was replaced with 5 ml of retrovirus solution supplemented with a final concentration of 8 µg/ml Polybrene (Sigma-Aldrich). Cells were then centrifuged for 1 h at $800 \times g$ and the retrovirus solution was replaced by fresh medium afterwards. Treatment was repeated after 10 hours and after 24 hours. On day 2 (24 h after the last treatment), cells were harvested and seeded in 10-cm dishes on mouse embryonic fibroblast (MEF) feeder cells, one well of the 6-well plate per dish. On day 3, medium was changed to iPSC medium adding 0.1% SB431542 (StemCell Technologies). Medium was changed every other day. For the feeder-dependent approach of the CytoTune(TM)-iPS 2.0 Sendai Reprogramming kit, on day −2 fibroblasts were seeded in increasing concentrations on two 6-well plates ($10–35 \times 10^4$ cells/well) with reprogramming fibroblast medium (RFM; DMEM, 10% ESC-qualified FBS (Thermo Fisher), 1% NEAA (Thermo Fisher), 0.1% 2-mercaptoethanol (Thermo Fisher)). On day 0, viral transduction was performed on 30–60% confluent cells using the Sendai virus-based reprogramming vectors containing the four Yamanaka factors, Oct3/4, Sox2, Klf4, and c-Myc[70]. The required virus amount was calculated using the given formula from the manufacturer's protocol and added to 1 ml of RFM replacing the old medium. 24 hours after transduction, medium was replaced by fresh RFM. Cells were then fed every other day. On day 5 after transduction, MEF feeder cells were thawed and seeded on a 6-well plate ($2.5 \times 10^5$ cells/well) as described in the manufacturer's protocol. On day 7 after transduction, fibroblasts were harvested, and seeded in different concentrations (5, 10, 20, 40, 80, $100 \times 10^3$ cells/well) onto the previously prepared 6-well plate with MEFs and cultured with RFM. On day 8, medium was replaced with iPSC medium (DMEM/F-12 (Thermo Fisher), 20% KOSR (Thermo Fisher), 1% NEAA, 0.1% 2-mercaptoethanol, 1% P/S, 0.04% bFGF (Thermo Fisher)) and then changed every day. About 21–28 days after transduction, iPSC colonies were emerging from the plate and approximately 50 colonies were picked and transferred to a single well of a 12-well plate coated with Matrigel (Corning) and cultured with mTeSR1 (StemCell Technologies). Medium was changed every other day; colonies were monitored regularly and expanded until ready for characterization. All iPSC lines were cultured in colonies in mTESR1 medium (StemCell Technologies) on Matrigel-coated dishes in 5% CO2 at 37 °C until they reached a confluency of 80–90% and then replated at a ratio of 1:3 to 1:10 using a self-made enzyme-free splitting buffer containing PBS, NaCl and EDTA. Spontaneously differentiated cells were manually removed under a stereomicroscope. iPSC lines used were regularly karyotyped. All cells were regularly tested for the presence of mycoplasma using the PCR Mycoplasm Test Kit (PromoKine) or the LookOut Mycoplasma PCR detection kit (Sigma-Aldrich).

### Ethical approval
Patients' fibroblasts[71–73] to generate M-ctrl, M-OS/het, M-OS/male, M-OS/maleR iPSCs were established in the Bristol Genetics Laboratory, following consent for further analysis and usage for research in an anonymized way was given by the family. Fibroblasts to generate J-ctrl, J2-ctrl, J-OS/het, J-OS/hom, J2-ctrl, the A-ctrl, A-OS/het, A-OS/hom, C1-male, and the KO-line iPSCs were taken at the University Medical Center in Mainz following approval by the local ethical committee (No. 4485). Consent for further analysis and usage for research in an anonymized way was given.

Animal experiments were conducted following German animal welfare legislation, and according to European (EU directive 2010/63/EU), national (TierSchG), and institutional guidelines.

### iPSC lines used in this study
A summary of all iPSC lines including information about the origin and genotype as well as for which experiments they were used is provided in Supplementary Data 1.

**The following female iPSC lines were used**. A female patient heterozygously carrying a 4-bp *MID1* deletion provided the M- fibroblasts, from which the M-ctrl and the M-OS/het iPSC lines were derived through clonal expansion. Two healthy female individuals provided fibroblasts termed A- and J- to derive the iPSC lines A-ctrl and J-ctrl (low *XIST*) and an additional clone, termed J2-ctrl (high *XIST*).

Using CRISPR/Cas9-mediated genome editing the 4-bp *MID1* mutation was introduced either heterozygously (J-OS/het, A-OS/het) or homozygously (J-OS/hom, A-OS/hom) into the J-ctrl and A-ctrl lines, respectively.

As additional control of a healthy individual, the hoik1 line from the HipSci feeder free panel (ECACC 77659901) was used for generating the snRNA-seq data of iPSC and brain organoids and used for RNA-FISH experiments.

**The following male iPSC lines were used.** The *MID1* KO-line[74] was generated through CIRSPR/Cas9-mediated knockout of the *MID1* gene in a male iPSC line derived from a healthy individual (C1-male). In addition, the patient-derived M-OS/male and the corresponding isogenic control M-OS/maleR, in which the 4-bp deletion in the *MID1* gene was corrected by CRISPR/Cas9 mediated genome editing, were employed.

## Clonal selection of iPSC lines
Following reprogramming into iPSC, individual clones derived from the M-, A- and J-fibroblasts, respectively, were picked and propagated for further analyses. Monoallelic expression of one X-chromosome in iPSCs was confirmed by using allele-specific PCR for the X-linked *MID1* gene (M-line) and by using a QUASEP assay for a heterozygous SNP located in the 3´untranslated region of the X-linked gene *ZNF185* (M-, A-, and J-line).

## Genome editing
For CRISPR/Cas9 genome editing iPSCs were electroporated using the Lonza 4D-NucleofectorTM X Unit. 800.000 single cells per transfection reaction were pelleted. Cells were resuspended in 100 µl electroporation buffer P3 (P3 Primary cell solution box, Lonza) plus 2.5 µg of the CAG-Cas9-Venus plasmid[75] (pU6-(BbsI)sgRNA_CAG-Cas9-venus-bpA was a gift from Ralf Kuehn (Addgene plasmid # 86986)) and 2.5 µg of gRNA-containing plasmid. The CAG-Cas9-Venus plasmid did not contain any gRNAs, but the gRNAs were provided by using the gRNA cloning vector[76] (gRNA_Cloning Vector was a gift from George Church (Addgene plasmid # 41824)). Cell-plasmid solution was transferred to the electroporation cuvette (Amaxa™ P3 primary cell 4D-Nucleofector™ X, Lonza) and electroporated using the program CB-150 of the Nucleofector. After electroporation 100 µl of RPMI (Thermo Fisher) was pipetted into the cuvette and incubated for 10 min at 37 °C and then transferred to a fresh well of a Matrigel coated 6-well plate with 2 ml of mTeSR1 supplemented with ROCK-inhibitor (10 µM, StemCell Technologies). Single GFP-positive cells were sorted in a 96-well plate with mTeSR plus CloneR (StemCell Technologies). 14 days after plating, colonies were transferred to a 12-well plate with mTeSR plus CloneR. A small volume (~10 µl) of the dissociated cells was used for DNA isolation by using Quick-DNA Microprep Kit (Zymo research). The DNA was amplified by PCR and sequenced via Sanger Sequencing. Clones carrying the desired mutation were selected for further expansion.

## 2D iPSC differentiation into neural progenitor cells (NPCs)
iPSCs were differentiated into NPCs with the PSC Neural Induction Medium (NIM) (Gibco)[77]. On day −1, iPSCs were washed with DMEM/F-12 and incubated with Collagenase IV (Thermo Fisher) for 20 min, followed by three wash steps with DMEM/F-12. Cells were scraped and resuspended with mTeSR1, transferred to ultra-low attachment plates and incubated for 24 h. On day 0, medium was replaced with neuronal medium (NM: DMEM/F-12, 1% N2-supplement (Thermo Fisher), 2% B27-supplement (Thermo Fisher), 1% P/S) and replaced every other day. On day 7, EBs were seeded on one well of a 6-well plate coated with poly-Ornithine/Laminin (Thermo Scientific). Medium was changed every other day. On day 14, rosette-like structures were visible inside the attached EBs and were transferred to a new well of a poly-Ornithine/Laminin-coated 6-well plate. On day 2 after replating, medium was replaced and supplemented with 0.1% FGF2 (Thermo Fisher). NPCs were expanded as needed or frozen in liquid nitrogen for long-term storage. M-ctrl NPCs for bulk RNAseq were generated according to this protocol. All other iPSC-to-NPC differentiations followed a similar but commercial alternative: on day −1, iPSCs were seeded on a Matrigel-coated 6-well plate. After 24 h, when cells reached a confluency of 15–25%, medium was changed to Neural Induction Medium (NIM: Neurobasal (Thermo Fisher), 2% Neural Induction Supplement (Thermo Fisher), 1% P/S), renewed with increasing volumes every other day to compensate for cell growth. On day 7 of neural induction cells were replated following the manufacturer's protocol in a Geltrex-coated (Thermo Scientific) 6-well plate (500 × 10³ cells/well) in Neural Expansion Medium (NEM: 49% Neurobasal, 49% Advanced DMEM (Thermo Scientific), 2% Neural Induction Supplement, 1% P/S) supplemented with ROCK-inhibitor (5 µM, StemCell Technologies). After 24 h, medium was exchanged with NEM and cells were monitored daily with medium changes every other day. When reaching confluency, cells were replated using TrypLE™ Express and cultured on poly-Ornithine/Laminin-coated dishes with NM supplemented with FGF2 (20 ng/ml).

## 2D differentiation of NPCs into neurons
NPCs were seeded at low confluence (50–100 × 10³ cells/well) onto poly-Ornithine/Laminin-coated cavities of a 6-well plate. 24 h after seeding, cells were washed twice with PBS and medium was replaced by NM+VitA (DMEM/F-12, 1% N2-supplement, 2% B27+VitA supplement (Thermo Fisher), 1% P/S). Cells were cultured in a humidified incubator at 37 °C and 8% $CO_2$. Every 3–4 days, fresh medium was added to the cells. When reaching a total volume of 10–12 ml per well, 20–50% of the medium was removed every time before adding fresh media. After 35 days, differentiated neurons were harvested as needed.

## Brain organoid formation
All iPSC lines were cultured in colonies in mTeSR Plus medium (StemCell Technologies) on Matrigel (Corning)-coated dishes in 5% $CO_2$ at 37 °C until they reached a confluency of 80–90%. Brain organoids were generated with slight modifications following the Lancaster protocol[78]. Accutase (Gibco) was used to generate single cell suspensions of cells. Following centrifugation, cells were resuspended in organoid formation medium supplied with 4 ng/ml of low bFGF (Peprotech) and 5 µM ROCK-inhibitor (StemCell Technologies). Organoid formation medium consisted of DMEM/F12 + GlutaMAX-I (Gibco), 20% KOSR (Gibco), 3% FBS (Gibco), 0.1 mM MEM-NEAA (Gibco), 0.1 mM 2-mercaptoethanol (Sigma-Aldrich). 12,000 cells in 150 µL organoid formation medium/well were reaggregated in low attachment 96-well plates (Corning) for at least 48 h. After 72 h half of the medium was replaced with 150 µl of new organoid formation medium without bFGF and ROCK-inhibitor. At day 5 neural induction medium consisting of DMEM/F12 + GlutaMAX-I (Gibco), 1% N2 supplement (Gibco), 0.1 mM MEM-NEAA (Gibco), and 1 µg/ml Heparin (Sigma-Aldrich) was added to the EBs in the 96-well plate to promote their growth and neural differentiation. Neural induction medium was changed every two days until day 12/13, when aggregates were transferred to undiluted Matrigel (Corning) droplets. The embedded organoids were transferred to a petri dish containing organoid differentiation medium without vitamin A. Organoid differentiation medium consisted of a 1:1 mix of DMEM/F12 + GlutaMAX-I (Gibco) and Neurobasal medium (Gibco), 0.5% N2 supplement (Gibco), 0.1 mM

MEM-NEAA (Gibco), 100 U/ml penicillin and 100 μg/ml streptomycin (Gibco), 1% B27 ± vitamin A supplement (Gibco), 0.025% insulin (Sigma-Aldrich), 0.035% 2-mercaptoethanol (Sigma-Aldrich). Three to four days later the medium was exchanged with organoid differentiation medium containing vitamin A and the plates were transferred to an orbital shaker set to 30 rpm inside the incubator. Medium was changed twice per week. For fixation, organoids were transferred from petri dishes to 1.5 ml tubes at day 30 and day 50. Organoids were washed with PBS and then fixed with 4% paraformaldehyde (PFA, Carl Roth) for 30 min. Time of PFA fixation was extended up to one hour depending on the size of the organoids. Afterwards, organoids were washed three times for 10 min with PBS and incubated in 30% sucrose (Sigma-Aldrich) in PBS for cryoprotection. For cryosectioning, organoids were embedded in Neg-50™ Frozen Section Medium (Thermo Fisher) on dry ice. Frozen organoids were cryosectioned in 30 μm sections using the Thermo Fisher Cryostar NX70 cryostat. Sections were placed on SuperFrost Plus™ Object Slides (Thermo Fisher) and stored at −20 °C until use.

### RNA-FISH procedure and imaging

To trace the allelic expression of individual X-chromosomal genes with a single cell resolution we employed the RNAScope technology (Bio-Techne) by following the manufacturer´s protocol with small adjustments. Cells seeded on coverslips were fixed when reaching 80–90% of confluency with 1 ml of freshly prepared PFA (4% in PBS) for 30 minutes at room temperature. After three washes with PBS, cells were dehydrated by incubation with increasing volumes of ethanol, 50%, 70%, and 100% for 1 min. 100% ethanol was replaced with fresh 100% ethanol, and cells were stored at +4 °C for up to 48 h or −20 °C for up to 6 months. Cells were rehydrated (70% and 50% ethanol for 1 min) followed by incubation with PBS for 10 min. For the pre-treatment procedures, coverslips were covered for 10 min with RNAScope Hydrogen Peroxide solution and further incubated for 10 min with RNAScope Protease III diluted 1:5 with PBS (cat. no. PN 322381). Hs-XIST-C2 probes (cat. no. 311231-C2) were mixed with Hs-MID1-O1-C3 (cat. no. 1224591-C3), Hs-GPM6B-intron-C3 (cat. no. 1201031-C3), Hs-NLGN4X-O1-C3 (cat. no. 1180201-C3), or Hs-SPIN3-Intron-C3 (cat. no. 1329651-C3) according to the manufacturer´s protocol. Coverslips were incubated with mixed probes for 2 h at 40 °C. Negative control samples were incubated with Probe Diluent (cat. no. 300041). Additional negative control samples were treated with RNase A (100 μg/ul, cat. no. R1253, Thermo Scientific) for 1 h at +37 °C prior probe incubation. The subsequent steps of amplification and signal detection were performed as described in the protocol. The Opal Dye 520 (Akoya) was used for Hs-XIST-C2 (1:750, excitation 494 nm, emission 525 nm), and Opal Dye 690 (Akoya) for Hs-MID1-O1-C3, Hs-GPM6B-intron-C3, Hs-NLGN4X-O1-C3 and Hs-SPIN3-Intron-C3 (1:750, excitation 676 nm, emission 694 nm) as fluorophores. Samples were mounted with ProLong Glass Antifade Mountant and ProLong Gold Antifade Mountant (Thermo Fisher). Fluorescence microscopy images were acquired using the VisiScope 5 Elements spinning disc confocal (Visitron Systems GmbH), built on a Ti-2E (Nikon) stand and equipped with a spinning disc unit (CSU-W1, 50 μm pinhole, Yokogawa), and controlled by the VisiView software (v7.0.0.7). Images were acquired with a 60x water-immersion (NA 1.2 Plan Apo lens) objective with an additional 2x magnification, and a sCMOS camera (BSI, Photometrics). BC43 spinning disk confocal (Oxford instruments) with a 100x oil-immersion (NA 1.45 Plan Apo lens) objective was also used for image acquisition. 3D stacks of images (with a voxel size of x 0.0568, y 0.0568, 100 nm for VisiScope, and x 0.0609, y 0.0609, 100 nm for BC43) were acquired for each sample.

### Quantification of RNA-FISH signals

At least 5 images for each experiment were acquired for the signal quantification. Images were analyzed using the Imaris Software (v10, Oxford instruments). Manual quantification was conducted after

setting the same threshold values for all pictures. *XIST*-positive nuclei were annotated as nuclei with a clear *XIST* cloud representing the inactive X-chromosome territory. Nuclei having at least one signal from the specific probe used were considered as positive nuclei for the specific probe. For the colocalization experiments between *XIST* signal and *MID1*, *GPM6B*, *NLGN4X*, and *SPIN3*, nuclei having a single probe signal non co-localizing with the *XIST* signal were considered as monoallelic nuclei, while nuclei having two probe signals, with one of them colocalizing with the *XIST* signal, were considered as biallelic nuclei. Pre-mRNA signals that were overlapping, adjacent to, or in close proximity (distant 2.5 μm) of the *XIST* signal were considered as co-localizing with the *XIST* cloud.

### Mouse embryonic tissue isolation and processing

Pregnant C57BL/6 mice at embryonic day E13.5 were anaesthetized with isoflurane and euthanized via cervical dislocation and embryos were harvested (E13.5). Subsequently, embryonic brains and skin tissue isolated from limb areas were dissected and fixed in freshly prepared 4% paraformaldehyde (PFA) overnight at 4 °C. Small portions of the tails were used for genotyping. Dehydration was performed by incubating the tissues in 50% ethanol for 1 h on the orbital shaker at 4 °C, followed by 70% and 100%. Tissues were stored at 4 °C in 100% ethanol prior further use.

### RNA-FISH on mouse embryonic tissue

The RNAScope technology (BioTechne) was employed by following the manufacturer´s protocol with small adjustments. Tissues were sliced in suspension, and a single thin slice was transferred into a well of a 96-well plate for the entire procedure. Tissues were rehydrated (70% and 50% ethanol for 1 min) followed by two incubation steps with PBS for 10 min. For the pre-treatment procedures, tissues were incubated for 10 min with RNAScope Hydrogen Peroxide solution and further incubated for 10 min with RNAScope Protease plus (322331) for 30 min. Mm-XIST-C1 probes (cat. no. 418281) were mixed with Mm-Mid1-O1-C3 (cat. no. 1802811-C3) according to the manufacturer´s protocol. Tissues were incubated with mixed probes for 2 h at 40 °C. Subsequent steps of amplification and signal detection were performed as suggested in the protocol. The Opal Dye 520 (Akoya) was used for Mm-XIST-C1 (1:750, excitation 494 nm, emission 525 nm), and Opal Dye 690 (Akoya) for Mm-Mid1-O1-C3 (1:750, excitation 676 nm, emission 694 nm) as fluorophores. Samples were mounted with Pro-Long Gold Antifade Mountant. Fluorescence microscopy images were acquired using the VisiScope 5 Elements spinning disc confocal (Visitron Systems GmbH), built on a Ti-2E (Nikon) stand and equipped with a spinning disc unit (CSU-W1, 50 μm pinhole, Yokogawa), and controlled by the VisiView software (v7.0.0.7). Images were acquired with a 60x water-immersion (NA 1.2 Plan Apo lens) objective with an additional 2x magnification, and a sCMOS camera (BSI, Photometrics). 3D stacks of images (with a voxel size of x 0.0568, y 0.0568, 100 nm) were acquired for each sample.

### Quantification of RNA-FISH signals in mouse embryonic tissues

At least three images for each experiment were acquired for the signal quantification. Images were analyzed using the Imaris Software (v10, Oxford instruments). Manual quantification was conducted after setting the same threshold values for all pictures. For each image, a set of nuclei with clear and well-defined borders was selected for further analysis. Xist-positive nuclei were annotated as nuclei with a clear Xist cloud representing the inactive X-chromosome territory. Nuclei having at least one signal from the Mid1 probe used were considered as positive Mid1 nuclei. For the colocalization experiments between Xist and Mid1, nuclei having a single probe signal non co-localizing with the Xist signal were considered as monoallelic nuclei, while nuclei having two probe signals, with one of them colocalizing with the Xist signal, were considered as biallelic nuclei. Pre-mRNA signals that were

overlapping, adjacent to, or in close proximity (distant 2.5 μm) of the Xist signal were considered as co-localizing with the Xist cloud.

## Immunohistochemistry

Cells on coverslips were fixed with 1 ml of PFA (4% in PBS) for 20 minutes, washed three times with PBS, and incubated with blocking solution (PBS, 5% BSA, 0.3% Triton) for 1 h. Afterwards, cells were incubated with primary antibodies diluted in blocking solution at 4 °C overnight. On the second day, cells were washed three times with PBS, 0.1% Triton for 10 min, followed by an incubation with secondary antibody diluted in blocking buffer for 1 h. After three wash steps with PBS, 0.3% Triton, coverslips were transferred to glass slides with 10 μl of mounting medium (Vectashield, 0.5% DAPI, Vector Laboratories). Organoid sections were post-fixed with 4% PFA for 15 minutes, washed three times for 5 min with 1 x PBS, and briefly washed with blocking solution containing 4% normal donkey serum (NDS, Sigma-Aldrich) and 0.25% Triton-X (Sigma-Aldrich) in 1 x PBS and subsequently incubated with blocking solution for at least one hour at RT. Tissue sections were incubated overnight at 4 °C with primary antibodies diluted in antibody solution containing 4% NDS (Sigma-Aldrich) and 0.1% Triton-X (Sigma-Aldrich) in 1 x PBS. Following 3 wash steps using PBS with 0.5% Triton-X (Sigma-Aldrich), secondary antibodies were diluted in antibody solution and incubated for one hour at RT. Finally, sections were washed three times for five minutes with PBS and PBS with 0.5% Triton-X was used for the last wash step. Slides were counterstained with DAPI 1:1000 in PBS for 5 min, washed with PBS and mounted using Aqua PolyMount (Polysciences, Inc.). Slides were kept in a humidified chamber in the dark during the entire staining procedure.

Antibodies used were selected according to the antibody validation reported by the distributing companies (information is provided in the reporting summary). Mouse (IgG1) anti-MAP2 (Sigma-Aldrich; M4403; HM-2; 139117; 1:300), rabbit anti-PAX6 (Biolegend; 901301; B277104; 1:300), rat anti-BrdU (Abcam; ab6326; BU1/75 (ICR1); GR3365969-9; 1:300), rabbit anti-H3K27me3 (Cell Signaling Technology; 9733S; C36B11; 27; 1:300), rabbit anti-Ki67 (Invitrogen; MA-14520; SP6; SI2454941R; 1:300), rabbit anti-p21 Waf1/Cip1 (Cell Signaling Technology; 2947; 12D1; 12; 1:300), human anti-PAX6 (Miltenyi Biotec; 130-107-582; 5170301048); 1:300), mouse (IgG1) anti-Phospho-Histone H3 (Ser10) (Cell Signaling Technology; 9706S; 6G3; 10; 1:300), rabbit anti-SOX2 (Abcam; ab137385; GR3313268-18; 1:300), mouse (IgG2b) anti-TUBB3 (Sigma; T8660; SDL.3D10; 127270; 1:300).The following secondary antibodies were used (1:500 dilution): goat anti-mouse IgG1 Alexa 488 (Thermo Fisher; cat.no. A21121; 2083196), goat anti-mouse IgG Alexa 488 (Thermo Fisher; cat.no. A11001; 2140660), goat anti-rabbit Alexa 488 (Thermo Fisher; cat.no. A11008; 2521157), goat anti-rabbit Cy3 (Thermo Fisher; cat.no. A10520; 2160048), goat anti-rat Alexa 555 (Thermo Fisher; cat.no. A10522; 2153107), goat anti-mouse IgG1 Alexa 555 (Thermo Fisher; cat.no. A21127; 2110847)), goat anti-Human Alexa 555 (Thermo Fisher; cat.no. A21433; 2150293), goat anti-rat Alexa 633 (Thermo Fisher; cat.no. A21094; 2087716), goat anti-rabbit Alexa 633 (Thermo Fisher; cat.no. A21070; 2079350), goat anti-mouse IgG1 Alexa 633 (Thermo Fisher; cat.no. A21126; 2128996), Phalloidin-Atto647 (Sigma; cat.no. 65906).

## Microscopy and image analysis

Epifluorescence pictures were taken using AxioObserver (Zeiss), or the EVOS™ M7000 Imaging System (Thermo Fisher). Z-resolved pictures were acquired at a LSM710 Confocal Microscope (Zeiss) or with the Apotome 2 (Zeiss) equipped with the Colibri5 light source (Zeiss). To quantify the VZLS area we measured for each organoid section, the total organoid area (excluding areas covered by cysts), and the area covered by VZLS using the selection tools in FIJI (v1.52-1.53). In R (v3.5.1-4.1.2) we then calculated the fraction of organoid area covered by VZLS area, averaged these values from different sections from the same organoids and normalized the resulting values to the average value of

the respective control organoids (J-ctrl for J-ctrl, J-OS/het and J-OS/hom; M-ctrl for M-ctrl and M-OS/het; M-OS/maleR for M-OS/maleR and M-OS/male) in each batch. To quantify the area covered by SOX2 and MAP2 in organoid sections we used epifluorescence pictures acquired with the EVOS M7000 and employed OpenCV (v4.4.0 – 4.5.1) in python (v3.9.1-3.9.10) for automated thresholding and counting of thresholded pixels. The neural area was determined as the number of pixels thresholded for either SOX2 or MAP2 and the fraction of SOX2/neural area and the fraction of MAP2/neural area was calculated using NumPy (v1.21.5) and pandas (v1.3.4).

To quantify PAX6 protein levels we used sections stained for SOX2 and PAX6 and considered the central single optical sections of each Apotome z-stack. We segmented the SOX2 positive cells using the cyto3 model in CellPose (v2.2.3)[79] and quantified the PAX6 protein levels in each segmented SOX2 positive cell employing measure.regionprops of the scikit-image package (v0.21.0) in python (v3.8.19). The correct for differences in staining intensity due to batch-to-batch variations we normalized each value to the mean PAX6 level in corresponding controls of each batch. Finally, we plotted the normalized mean intensity of PAX6 vs the distance to the apical side for each SOX2 positive cell within a VZLS.

In a very similar approach, we quantified p21 protein levels in PAX6 positive cells, however this time the segmentation was done on PAX6 and both p21 as well as PAX6 levels were measured in segmented cells. To segment pH3 positive cells we used celldetection (v0.4.9).

## Western blot analysis

Protein lysates were generated from cell pellets using Magic Mix (48% urea, 15 mM Tris pH7.5, 8.7% Glycerin, 1% SDS, 143 mM 2-mercaptoethanol) containing protease and phosphatase inhibitors (cOmplete Tablets easypack, PhosSTOP easypack, Roche) and transferred to a QIAshredder column (Qiagen). After centrifugation at 13,500 g for 2 min, the solution was transferred into a fresh tube and frozen at −80 °C until needed. SDS gel-electrophoresis was used to separate proteins by their size. Proteins were transferred to a PVDF-membrane by using the Trans Blot Turbo Transfer Pack (Bio-Rad). Membranes were incubated for 24 h with blocking buffer (PBS, 0.1% Tween, 5% milk), followed by 24 h incubation with primary antibody diluted in blocking buffer. Membranes were washed 3 times for 10 min with PBS-T (PBS, 0.1% Tween), incubated for 1 h with secondary antibody diluted in blocking buffer, and washed 3 times for 10 min with PBS-T. Membranes were exposed using the Western Lightning Plus-ECL (Perkin Elmer) and imaging was performed by using ChemiDoc Imaging System (Bio-Rad). Images were prepared and analyzed using the Image Lab software (Bio-Rad). Mouse monoclonal anti-β-ACTIN (Sigma; A2066-200UL; 1:2000), rabbit polyclonal anti-MID1 C-terminal (Novus, NBP1-26612; 1:500), rabbit anti-MID1 N-terminal (courtesy of Dr. Sybille Krauss; 1:500)[80].

## qRT-PCR

Total RNA was extracted with the RNeasy Mini Kit (Qiagen) or the High Pure RNA Isolation Kit (Roche). Samples were stored at −80 °C until use. The RNA concentration was measured using a Nanodrop (PeqLab) or Qubit4 (Thermo Fisher). cDNA was generated starting from 125 ng up to 500 ng of total RNA with the Maxima First Strand cDNA Synthesis Kit (Thermo Fisher) or the PrimeScript RT Master Mix (Takara). In each individual experiment, equal amounts of RNA were used for the generation of cDNA. The primers used are listed in Supplementary Data 5. For qRT-PCR analysis, all samples were run in triplicates each with a reaction volume of 10 μL, using the QuantiFast SYBR® Green PCR Kit (Thermo Fisher) or TB Green Premix Ex Taq II (Tli RnaseH Plus) Kit (Takara), 1 μM primers and 1 μL of cDNA. The reaction was performed in a QuantStudio 6 Flex Real-Time PCR System (Thermo Fisher) or the StepOne Plus Real-Time PCR System (Thermo Fisher) using the following amplification parameters: 5 min at 95 °C, 40 cycles of 10 s at

95 °C and 1 min at 60 °C. Data were analyzed using the ΔΔCT method as previously described[81]; expression levels were obtained normalizing each sample to the endogenous *GAPDH* control.

## Allele-specific RT-PCR

Specific primers were generated to bind to the wildtype or to the mutant (4-bp deletion) *MID1* allele (Supplementary Data 5). For each sample, two reactions were performed, one with the reverse wildtype primer, and one with the reverse mutant primer, both using the same forward primer. After an initial denaturation step (95 °C for 2 min), 35 cycles of denaturation step (95 °C for 30 s), primer annealing step (70 °C for 30 s), and elongation step (72 °C for 30 s) were followed by a final elongation step (72 °C for 5 min). The PCR products were analyzed by separation on a 1.5% agarose gel with EtBr.

## Quantification of allele-specific expression by pyrosequencing (QUASEP)

Amplification and sequencing primers (Supplementary Data 5) were designed by using the PyroMark Assay Design Software (v2.0, Qiagen) for the region of interest, with one of the amplification primers being biotinylated. PCR reactions were performed for each sample with the amplification primers using 60 °C as annealing temperature. Each sample was measured in triplicates. The PCR product was then used for pyrosequencing following the standard protocol of the PyroMark Gold Q96 Reagents kit (Qiagen). A sequencing cartridge was prepared with all four nucleotides, the enzyme, and the substrate mix for sequencing, with the volumes calculated by the Pyro Q CpG (Qiagen) software. Each sample was measured in triplicates. The results were analyzed using the same software.

## Bulk RNA-sequencing

**M-ctrl and J-ctrl lines**. The library for RNA-seq experiments was prepared from 5 ng of total RNA using the Ovation® Solo RNA-Seq Library Preparation Kit (Tecan) following the manufacturer's instructions. The library was denatured for sequencing by mixing 5 µl of the 4 nM library with 5 µl of NaOH (0.2 M). After incubating for 5 min, 5 µl of Tris buffer (200 nM Tris-HCl, pH 7) were added. Finally, 985 µl of prechilled HT1 were added. The sequencing run on a NextSeq 500/550 was performed as a paired-end run with 2 times 76 cycles and an expected output of 50 million reads per sample for allele-specific analysis.

**A-ctrl line**. NGS library prep was performed with Illumina's Stranded Total RNA Prep Ligation with Ribo-Zero Plus Kit following Stranded Total RNA Prep Ligation with Ribo-Zero Plus ReferenceGuide (Document # 1000000124514 v02 April 2021). Libraries were prepared with a starting amount of 613 ng and amplified in 10 PCR cycles. Two post PCR purification steps were performed to exclude residual primer and adapter dimers. Libraries were profiled in a DNA 1000 chip on a 2100 Bioanalyzer (Agilent technologies) and quantified using the Qubit dsDNA HS Assay Kit, in a Qubit 4.0 Fluorometer (Invitrogen by Thermo Fisher Scientific). All samples were pooled in equimolar ratio and sequenced on 1 NextSeq2000 P3 flow cell, PE for 2 × 151 cycles plus 2 × 10 cycles for the dual index read and 1 dark cycle upfront R1 and Read 2.

## Bulk RNA-seq data pre-processing

**M-ctrl and J-ctrl lines**. Sequencing reads were demultiplexed and base call (BCL) files were converted into Fastq files using bcl2fastq conversion software (v2.17.1.14, Illumina). Sequence adapters were trimmed and reads shorter than 6 bp were removed from further analyses using Cutadapt (v0.18). Quality control checks were performed on the trimmed data with FastQC (v0.11.7). Read mapping of the trimmed data against the human reference genome and transcriptome (hg19) was conducted using the STAR aligner (v2.5.3)[82]. PCR duplicates were removed from the mapped reads using the python script nudup.py

(v2.3) provided from NuGen (https://tecangenomics.github.io/nudup/).

**A-ctrl line**. Adapter and quality trimming were performed with the BBDuk tool from BBMap (v38.86). Quality control was facilitated using FastQC (v0.11.9). Reads were then aligned against the human reference genome (hg19) using STAR (v2.7.10b). A count table was obtained with featureCounts from the Subread package (v2.0.1).

## Allele-specific expression analysis (ASE) of bulk RNA-seq data

For ASE, all bulk RNA-seq data from the M-ctrl, J-ctrl and A-ctrl lines were processed with the NVIDIA Clara Parabricks Pipeline (v3.5) (https://www.nvidia.com/en-us/clara/genomics/). The built-in RNA pipeline rna_gatk was employed to align the fastq-files to the hg19 reference genome with the built-in STAR[82] aligner. After coordinate sorting and marking duplicates of the resulting BAM file, a base recalibration step was performed before variant calling with built-in gatk Haplotype Caller[83,84]. Analyses were restricted to the X-chromosome by defining -L chrX. For joint genotyping, a genomic database was built with built-in gatk GenomicsDBImport including all variants from all samples and subsequent genotyping of the X-chromosomal variants was performed with built-in gatk GenotypeGVCFs. Gene names were annotated by the Ensembl Variant Effect Predictor[85] before transforming the resulting VCF file to a table format with gatk VariantsToTable for smaller file size and further downstream analysis in R. We used R base functions to remove all variants that did not intersect known genes, multi- and monoallelic variants, variants with a quality score <100 as well as intermediate and very long indels (>50 bp). Then we calculated the allelic ratio for each variant site by dividing the number of reads mapping to the reference allele (No. Ref) by the total number of reads covering the variant site (No. Total). For further processing, we set a threshold of at least 5 reads in all samples to consider a variant (SNP or indel) to be expressed. We then calculated average allelic ratios for each cell type (iPSCs, NPCs, and neurons) in the sequenced M-ctrl, J-ctrl and A-ctrl lines, respectively, by summing up all reads mapping to the reference allele across all replicate samples from the corresponding cell type and cell line and dividing this number by the total number of reads covering the variant site. This analysis was performed only including variant sites that were covered by at least 20 reads on the group level. To detect reactivated genes which are characterized by monoallelic expression in iPSCs but biallelic expression in NPCs and/or neurons, we first identified all variant sites with an allelic ratio <0.025 or >0.975 in iPSCs. The allelic ratio was converted to an estimate of the probability of expression $\hat{p}_{Xi}$ from the inactive chromosome Xi using the following formula (1):

$$\hat{p}_{Xi} = \begin{cases} \frac{No.Ref}{No.total} & \text{if } No.Ref < No.Alt \\ 1 - \frac{No.Ref}{No.total} & \text{otherwise} \end{cases} \quad (1)$$

We then employed one-sided binomial tests to investigate if $\hat{p}_{Xi}$ is significantly greater than 0.025, indicating statistically significant expression from the inactive X-chromosome. *P*-values were adjusted for multiple comparisons using the Benjamini-Hochberg method implemented in the stats R package (v4.0.2). Results with an adjusted *P* < 0.01 were considered statistically significant, indicative of a reactivated variant site. Variant sites were considered as escapees of XCI if $\hat{p}_{Xi}$ is significantly greater than 0.1 in all cell types (iPSCs, NPCs, and neurons). To detect late-silenced ASE sites, we first identified all variants with $\hat{p}_{Xi}$ < 0.025 in NPCs and neurons and then used one-sided binomial tests to examine if Xi expression was significantly greater than 0.025 in iPSCs.

## Gene-wide estimates of biallelic expression

For genes that intersected only one ASE variant site, we used the Xi expression for the respective variant as an overall estimate of biallelic

expression for the whole gene. If a gene contained more than one type of biallelically expressed variants, the gene was assigned to a single category (reactivated, escapee or late-silenced) following a predefined set of rules: (1) the gene was assigned to the category with the highest number of biallelic variants (2) if a gene covered the same number of biallelic variants from different categories, then SNPs were considered as more reliable than indels, and variants of the same type with a higher coverage were preferred. After assigning genes to a unique category, we used the metaprop function from the meta R package (v4.18.1) to calculate gene-level estimates of biallelic expression from values for individual variants by using an inverse-variance method with logit-transformed proportions of Xi expression. 99% confidence intervals of biallelic expression based on a normal approximation were also obtained with the metaprop function.

## Manual curation of biallelically expressed variant sites

All predicted biallelic variant sites were subjected to manual inspection. 22 variants from M-ctrl, 14 variants from J-ctrl and 59 variant sites from the A-ctrl line were flagged as false positives. In most cases the predicted biallelic expression for these sites was due to all but one replicate expressing the reference allele and only one the alternative (or vice versa). These positions were then excluded from the analysis.

## X-chromosome ideograms

Ideograms showing the cytogenetic location of reactivated, escape and late-silenced genes were produced with the karyoploteR package (v1.4.1) based on the hg19 genome assembly for human and mm10 for the mouse. Gene coordinates were obtained from the Ensembl/Gencode annotation.

## Overlap with known escape genes in humans

The overlap of reactivated, full escape and late-silenced genes in this study with known escape genes in humans was obtained using escape status assignment from two previous studies[8,13]. To calculate an expected value, we performed 1000 random sampling iterations by drawing the number of genes in each category (reactivated: $n = 95$; full escapees: $n = 82$; late-silenced: $n = 29$) from all X-linked genes in the Ensembl annotation and determining the overlap with known escape genes. The enrichment was calculated by dividing the observed overlap by the mean value for the expected overlap from the 1000 sampling iterations. The significance of the observed overlap was obtained using the cumulative distribution function of the normal distribution.

## Overlap with known escape genes in the mouse

To assess the escape status of orthologues of reactivated, full escape and late-silenced genes in the mouse, we utilized a recently published study by Hauth et al.[32], which described constitutive, variable and NPC-specific escape genes. Enrichment of previously reported mouse escapees for our reactivated, full-escapee and late-silenced genes was calculated as described above for the overlap with previously reported escapees in humans.

## Enrichment of NDD-associated genes

To obtain the overlap of reactivated, full-escape and late-silenced genes with NDD genes, we used the recently published GeneTrek database containing high-confidence NDD genes[16]. Then, we calculated the expected overlap of biallelically expressed genes with NDD genes by randomly sampling the 513 X-chromosomal genes present in the GeneTrek database that are also expressed in the NPCs and neurons in our study (>10 reads in each sample following DESeq2 (v.1.42.1) normalization of the data). In each one of 1000 iterations we randomly drew the number of genes present in the respective category (reactivated: $n = 95$, late-silenced: $n = 29$, escapee: $n = 82$) and determined the number of NDD associated genes in each

iteration. The mean value over all iterations was determined as the expected number of NDD genes and the enrichment was calculated as the ratio between observed/expected number of NDD genes in each class. The significance of the observed overlap in each class was determined by the cumulative distribution function of the normal distribution.

## Pairwise distance analysis of biallelically expressed genes

To investigate if different types of biallelically expressed genes cluster together on the X-chromosome, we calculated the mean pairwise distance between transcription start sites for reactivated, escapee, and late-silenced genes, respectively. We tested the probability of observing a lower mean pairwise distance by chance by comparing the actual distance to a background distribution of pairwise distances for 1000 sets of randomly selected X-linked genes which were expressed in NPCs and neurons in our bulk RNA-seq data (>10 reads in each sample following DESeq2 (v1.42.1) normalization of the data). The $P$-value of observing a lower mean pairwise distance by chance was obtained with the cumulative distribution function of the normal distribution.

## Protein–protein interaction network of reactivated genes

To construct a protein–protein interaction (PPI) network of reactivated genes, we employed the data provided by the human binary reference interactome (HuRi)[86] available at http://www.interactome-atlas.org/. The 54 protein-coding reactivated genes that were included in the HuRi portal, were used to create the network. To construct a differentiation-specific PPI, we filtered for genes that were expressed in all NPC and neuron samples in our bulk RNAseq data (>10 reads in each sample following DESeq2 (v1.42.1) normalization). We employed the igraph package (v1.2.6) in R to visualize the network. GO term analysis of the PPI network of reactivated genes was performed with the clusterProfiler package (v3.16.1) using the default settings of clusterProfiler, that is an upper-tail hypergeometric test with Benjamini Hochberg correction for multiple comparisons. GO terms with an adjusted $P$-value below 0.05 were considered significant. Redundant GO terms were combined using the simplify function.

## Chromatin state analysis of biallelic genes

To investigate the association between biallelic expression and chromatin state, we employed the 15-state chromatin model from the Roadmap Epigenomics Consortium[30] using the female (sample E082) and male (sample E081) fetal brain samples. The annotation files for the epigenomes were downloaded from https://egg2.wustl.edu/roadmap/web_portal/chr_state_learning.html. We used the ChromDiff tool[87] to estimate the percentage of each gene body corresponding to each of the 15 chromatin states in the two epigenome samples. We then compared the chromatin state profiles of the 83 reactivated genes, the 63 full-escapees and 20 late-silenced genes that were also included in the GENCODE annotation files distributed with the ChromDiff tool against the profiles of X-chromosomal genes expressed in NPCs and neurons (>10 normalized reads in all samples) but not showing biallelic expression in our assay and not being previously reported as escapee genes[8,13] (inactive genes, $n = 128$). We then performed one-sided Wilcoxon rank sum tests to investigate the enrichment of epigenomic states in reactivated, full-escapee and late-silenced compared to inactive genes separately in the female and male fetal brain samples. $P$-values were adjusted for multiple comparisons with the Benjamini-Hochberg method and an epigenomic state was considered enriched in the respective gene category if the adjusted $P$-value was <0.05.

## Meta plots of histone modification signal in female fetal brain tissue

Meta plots for H3K36me3, H3K9me3, H3K27me3 and H3K4me1 signal for reactivated, full-escape, late-silenced and inactive genes in female

fetal brain tissue from 17-week-old embryos were produced using the computeMatrix and plotProfile functions from deepTools v3.5.4. bigWig files with fold change over control were obtained from ENCODE, reference epigenome ENCSR189GMC.

## Sex-biased expression of reactivated, escapee, and late-silenced genes

Information about sex biases in the expression of the biallelic and inactive genes predicted here was extracted from the study by Oliva[13] and colleagues which investigated systematic differences of gene expression between males and females across 44 human tissues from GTEx V8. We compared the distributions of the number of tissues with a female bias in each of our gene categories compared to inactive genes using Fisher's exact test.

## Disease-ontology enrichment analysis

Disease ontology enrichment analysis for reactivated genes was performed using the DOSE R package v3.28.2 with an adjusted p-value cutoff of 0.05. X-linked genes from the Ensembl annotation were used as the background universe for this analysis.

## Single cell RNA-seq data generation

For the scRNA-seq experiment, organoids were dissociated using the Neural Tissue Dissociation Kit P (Miltenyi Biotec). Briefly, selected organoids were cut into smaller pieces, washed with medium, and three times for 5 min with 1xPBS. Organoid pieces were transferred to a tube containing the enzyme mix P (according to the manufacturer's protocol) and incubated at 37 °C for 10 min. Pieces were then triturated gently with a 1000p pipette tip and incubated for another 10 min at 37 °C in the presence of enzyme mix A (according to the manufacturer's protocol). Pieces were then triturated gently with a 1000p and a 200p pipette tips and incubated for 5 min at 37 °C. Cell suspension was filtered with a 30 µm filter (Miltenyi Biotec) and centrifuged at $300 \times g$ for 5 min. After a second filtration step with a 20 µm filter (Miltenyi Biotec) and subsequent centrifugation as described above cell pellet was resuspended in 100 µl 1xPBS (without $Ca^{2+}$ and $Mg^{2+}$). Cells were counted and tested for viability with Trypan Blue and the automated cell counter Countess (Thermo Fisher). Cells were diluted to an appropriate concentration to obtain approximately 5000 cells per lane on a 10X Next GEM chip G v3. Libraries were constructed according to the Chromium Single Cell 3´ Reagent Kit User Guide (v3 Chemistry Dual Index) of 10x genomics and sequenced on an Illumina NovaSeq 6000.

## Single nucleus RNA-seq data generation

To generate single cell solutions, organoids were dissociated using the Neural Tissue Dissociation Kit P as described above, while iPSCs were detached and singularized using Accutase. Cells were counted and tested for viability with Trypan Blue and the automated cell counter Countess. Nuclei were isolated following 10X Genomics CG000124 Rev F – Nuclei Isolation from Cell Suspensions & Tissues for Single Cell ENA Sequencing with slight adaptations. Briefly, $2.5 \times 10^6$ cells were pelleted at 400 g for 5 min n at 4 °C. Cells were lysed using 0.025% Nonidet P40 Substitute (Millipore-Sigma), 10 mM NaCl (Millipore-Sigma), 3 mM $MgCl_2$ (Millipore-Sigma) in Ambion Nuclease-Free water (Invitrogen) for 4 min. Lysis was stopped using Nuclei wash (1xPBS (without $Ca^{2+}$ and $Mg^{2+}$), 1%BSA (Carl Roth), 0.2 U/µL RNase Inhibitor (Millipore-Sigma)) and centrifuged at $500 \times g$ for 10 min at 4 °C. Nuclei were washed twice with Nuclei wash. Nuclei were counted and tested for successful cell lysis with Trypan Blue and the automated cell counter Countess. Two cell lines were multiplexed leveraging SNPs for later demultiplexing. Nuclei were diluted to an appropriate concentration to obtain approximately 10,000 cells per lane on a 10X Next GEM chip G v3. Libraries were constructed according to the protocol of 10X Genomics and sequenced on an NovaSeq 6000.

## Single nucleus and single cell RNA-seq data preprocessing, clustering, visualization

The functions count and aggr of the Cell Ranger software (10x Genomics, v4.0.0 – 6.0.2) were used for aligning to the GRCh38 reference genome and for sequencing depth normalization. The multiplexed single nucleus RNA-seq data was demultiplexed using souporcell[88] (v2.5) using default settings. Scanpy[89] (v1.8.1) was used for further preprocessing, clustering, embedding, and visualization. Cells were excluded in which we detected less than 500 genes, less than 3000 UMI counts, more than 50,000 counts, fraction of the transcriptome of less than 1% or more than 10% mitochondrial genes. Moreover, genes expressed in less than 5 cells were excluded from further analysis. Doublets were identified with Scrublet[90] (v0.2.3) and filtered out. After normalization and log transformation, highly variable genes were determined using the default settings in Scanpy. Cell cycle scores, percentage of mitochondrial genes, and number of detected UMIs were regressed out to reduce their confounding effects. PCA analysis with the arpack wrapper in SciPy (v1.7.0) followed by determining the 15 closest neighbors in the top 20 PCs, batch integration using bbknn's (v1.4.0)[91] Euclidian metric and force-directed graph embedding utilizing ForceAtlas2[92] provided by the python package fa2 (v3.5) was used for embedding the transcriptomes. Clustering was performed with Scanpy's python implementation of the Leiden[93] algorithm (v0.8.7). For a coarse-grained mapping of clusters, their connectivity and lineages we applied the Scanpy implementation of the partition-based graph abstraction (PAGA)[89].

## Single nucleus and single cell RNA-seq RNA velocity and pseudotime estimation

To determine spliced and unspliced transcripts, the Cell ranger produced BAM files were sorted by the cell barcode with samtools (v1.10) and counted with velocyto (v0.17)[94]. RNA velocity estimation was then done with scVelo (v0.2.3)[95]. In detail, moments were calculated with the 'connectivities' mode on the top 50 PCs and 10 closest neighbors. After recovering the velocity dynamics, latent time was calculated, a measure for the developmental time (pseudotime) exclusively depending on transcriptional dynamics. RNA velocity was then computed, using latent time, differential kinetics and highly variable genes with the stochastic model.

## Differential gene expression and GO term analysis in the single cell RNA-seq data

Differentially expressed genes in NPC were determined between J-ctrl and J-OS/hom in the clusters 0, 1, 2, 3, 4, 5, and 6 of the neural lineages embedding using the rank_genes_groups function of Scanpy and testing for significance with the Wilcoxon rank sum test. Genes with an adjusted $P < 0.01$ and an absolute log2 fold change $>0.25$ were considered differentially expressed. GO analysis was done with the R (v4.1.2) package TopGO (v2.44.0) considering genes expressed in NPCs as background and excluding GO terms with less than 5 annotated genes from the analysis. The default weight01 algorithm was applied to determine the GO enrichment. Only GO terms in which more than 3 differentially expressed genes were found and with a $P$-value $< 0.05$ as determined by Fisher's exact test were considered.

## Transcriptional deviation

To determine the transcriptional deviation from control we determined for each cluster the differentially expressed genes between J-ctrl and J-OS/hom as well as M-ctrl and M-OS/het with $P < 0.01$ and an absolute log2 fold change $>2$ (excluding cluster 11 due to the lack of cells from the J-OS/hom condition). For each differentially expressed gene we calculated the average expression in both controls (M-ctrl and J-ctrl) in this cluster. For each differentially expressed gene in each cell of the cluster we determined the fold change over the average expression of this gene in the corresponding control. For upregulated

genes: cell$^{gene}$/(average ctrl$^{gene}$); for downregulated genes: 1/(cell$^{gene}$/(average ctrl$^{gene}$)). In each cell of the cluster, we determined the mean fold change of all cluster-specific differentially expressed genes as a measure for the transcriptional deviation. To determine the cluster-specific transcriptional deviation, we calculated the mean transcriptional deviation of all cells of that cluster in a cell line-specific manner. To resolve the temporal pattern of transcriptional deviation, the single cell values of transcriptional deviation were plotted against the RNA velocity based pseudotime estimation (latent time). The *MID1* expression levels as well as the ratio of *SOX2/SNAP25* expression were calculated by binning the pseudotime in 50 equally sized bins and calculating the mean values in J-ctrl cells.

### Processing and analysis of published data of developing human spinal cord

The raw fastq files from the human developing spinal cord single cell RNA-seq experiments in the Rayon dataset[34] were downloaded using fasterq-dump (v3.0.0). Processing of the raw reads was similar to the organoid single cell RNAseq data described earlier. In brief, we used the 10x genomics cellranger (v.7.1.0-8.0.0) functions, count and aggr to map the raw reads to the human reference genome GRCh38 and for sequencing depth normalization and then employed Scanpy[89] (v1.8.2) for preprocessing, clustering, embedding, and visualization. Cells were excluded in which we detected less than 500 genes, less than 1000 UMI counts, more than 20,000 counts, fraction of the transcriptome of less than 1% or more than 10% mitochondrial genes. Moreover, genes expressed in less than 5 cells were excluded from further analysis. Female samples were selected and further analyzed as described above for the organoid single cell RNA-seq data. The developmental pseudotime was estimated using the diffusion pseudotime[96] implementation in Scanpy. The *POU5F1, NANOG, SOX2, NES, DCX* and *SNAP25* expression levels were calculated by binning the pseudotime in 40 equally sized bins and calculating the mean expression value of all cells in each pseudotime bin.

### SNP calling and variant counting in sc and snRNA-seq data

To detect expressed heterozygous SNPs on the X-chromosome we pseudo-bulked the sequencing data of each sample and used Free-Bayes (v1.3.5) for detection. The bam files generated by cellranger for each sample were used to detect intragenic heterozygous variants with a PHRED quality score of at least 20 for each allele. Variants with a sequencing depth normalized threshold of at least 50 reads per $5 \times 10^8$ total reads were considered for further analysis in the single cell RNA-seq data and 2 reads per $5 \times 10^8$ total reads for the single nucleus RNA-seq data. The information on X-chromosomal variants from each line were then used for allele-specific alignments with the WASP[97] implementation in the STAR aligner (v2.7.8a)[82]. Using NumPy (v1.21.4) and pandas (v1.3.4) in python (v 3.9.7), the number of UMIs for each variant in each cell was determined. We then divided the number of UMIs of the most abundant allele by the total number of UMIs for that site (monoallelic=1, biallelic with 50% transcripts from each allele=0.5) and calculate the natural logarithm of this ratio as a measure for the allelic expression of a given gene in a given cell (monoallelic=0, biallelic with 50% transcripts from each allele=0.69). To calculate the degree of general biallelic expression from the X-chromosome in a cell, the values of biallelic expression for all biallelically expressed X-chromosomal genes were summed up. Pseudoautosomal genes were assigned using the HUGO gene nomenclature annotation for PAR1 and PAR2.

### Statistics and reproducibility

Data were statistically analyzed with GraphPad Prism, R or Microsoft Excel using statistical tests indicated throughout the manuscript. At least 3 biological replicas were used for every experiment, unless otherwise stated. No statistical methods were used to predetermine sample size. The investigators were not blinded to allocation and outcome analysis. The experiments were not randomized.

### Reporting summary

Further information on research design is available in the Nature Portfolio Reporting Summary linked to this article.

## Data availability

The bulk RNAseq data is deposited in the Sequence Read Archive (SRA) under the BioProject PRJNA819272. The FASTQ files of the single nucleus and single cellRNA-seq data are deposited in the European Nucleotide Archive (ENA) at EMBL-EBI under accession number PRJEB96835. Source data are provided as a Source Data file with this paper. Source data are provided with this paper.

## Code availability

All computational tools used in this study are publicly available.

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

## Acknowledgements

We thank Dandan Han for help with library preparation, the Helmholtz NGS core facility for sequencing, Magdalena Götz (BMC LMU Munich) and Michael Wegner (FAU Erlangen-Nürnberg) for sharing equipment and lab space, and Claudia Keller-Valsecchi (IMB Mainz) for valuable comments on the manuscript. We also thank the Genomics and Microscopy core facilities of the IMB Mainz. This work was supported by grants from the German Research Foundation (BE 4182/7-1, project number 2796480558), core funding to the Francis Crick Institute from Cancer Research UK, The Medical Research Council and the Wellcome Trust (FC001002), an ERA-NET Neuron grant (Brain4Sight, 01EW2202), and the ReALity initiative of the University Mainz to B.B., by the CRC 1080 (project number 402386039) and the German Research Foundation (SCHW 829/7-1 and BE 41882/7-1) and the ReALity initiative of the University Mainz to Su.S.; the German Research Foundation (KA3125/2-1; GRK2162/2 TP C2), Schram foundation (T287/29577/2017) to M.K.; the Bavarian State Ministry of Sciences, Research, and the Arts (ForInter; F.2-F2412.30/1/24) to M.K. and S.F.; the German Research Foundation (project number 460333672, subproject A04) to M.K. and S.F.; Interdisciplinary Center for Clinical Research (IZKF) at the University Hospital of the FAU Erlangen-Nürnberg to M.K. (Jochen-Kalden funding program N7) and S.F. (E32).

## Author contributions

B.B., Su.S., S.F., and M.K. conceived the study and designed the experiments. S.K., M.B., E.W., J.K., V.E., H.M.B.I., and A.S. generated and engineered iPSC lines. P.L. provided patient fibroblasts of the M-lines. B.W. and J.W. provided initial training for iPSC generation and cell lines. S.K., M.B., performed the 2D differentiation assays, QUASEP, western blot and (q)RT-PCR and analyzed the data. M.B. together with N.B. performed and together with C.F. and Sa.F. analyzed the RNA-FISH

experiments, co-supervised by M.F.B with input from P.B. Je. W. co-supervised S.K., M.B., and S.K. processed the bulk RNA-seq samples. S.D. preprocessed the data and performed variant calling while H.T. performed the bioinformatic analyses under the supervision of S.G. and with input from K.L. E.C. performed iPSC differentiation experiments under supervision of N.E. M.B. and A.M.S., supervised by M.F.B. conducted the RNA-FISH experiments in the mouse embryos. R.M., E.G., Sa.F., F.F., L.B., H.B., and B.L. conducted the brain organoid experiments and performed the cellular and molecular phenotyping of the ctrl and patient organoids. The authors S.K., R.M., and E.G. contributed equally. All organoid-related data were analyzed together with S.F. and M.K. The sc- and sn-RNAseq data was generated by S.F., E.G., and Sa.F., processed, and analyzed by S.F. and M.K. S.F. performed the analyses of the human spinal cord data. B.B., Su.S., S.F., and M.K. wrote the manuscript, with all authors contributing corrections and comments.

## Funding

## Competing interests
The authors declare no competing interests.

## Additional information

S. Schweiger, S. Falk or M. Karow.

[1]Institute of Human Genetics, University Medical Center of the Johannes Gutenberg University Mainz, Mainz, Germany. [2]Institute of Biochemistry, Friedrich-Alexander-Universität Erlangen-Nürnberg, Erlangen, Germany. [3]Institute of Physiological Chemistry, University Medical Center Mainz, Mainz, Germany. [4]Leibniz Institute for Resilience Research, Mainz, Germany. [5]Department of Biology, Johannes Gutenberg University Mainz, Mainz, Germany. [6]Division of Child Neurology and Metabolic Medicine, Center for Child and Adolescent Medicine, University Hospital Heidelberg, Heidelberg, Germany. [7]Institute of Human Genetics, Heidelberg University, Heidelberg, Germany. [8]Department of Industrial Engineering, Università degli studi di Padova, Padova, Italy. [9]Department of Molecular Neurology, University Hospital Erlangen, Friedrich-Alexander-Universität Erlangen-Nürnberg, Erlangen, Germany. [10]Department of Stem Cell Biology, Friedrich-Alexander-Universität Erlangen-Nürnberg, Erlangen, Germany. [11]Center of Rare Diseases Erlangen (ZSEER), Friedrich-Alexander-Universität Erlangen-Nürnberg, Erlangen, Germany. [12]Institute of Molecular Biology (IMB), Mainz, Germany. [13]Centre for Academic Child Health, University of Bristol, Bristol, UK. [14]Institute for Quantitative and Computational Biosciences (IQCB), Johannes Gutenberg University Mainz, Mainz, Germany. [15]Institute of Psychiatry, Psychology &Neuroscience, Centre for Developmental Neurobiology, King´s College London, London, UK. [16]MRC Centre for Neurodevelopmental Disorders, King´s College London, London, UK. [17]The Francis Crick Institute, London, UK. [18]Focus Program Translational Neuroscience, Johannes Gutenberg University Mainz, Mainz, Germany. [19]These authors contributed equally: M. Bertin, H. Todorov, S. Frank. ✉e-mail: schweigs@uni-mainz.de; sven.falk@fau.de; marisa.karow@fau.de

