## [Transparent Peer Review file · Nature Communications]

Dynamic allele usage of X-linked genes ameliorates neurodevelopmental disease phenotypes in brain organoids

Corresponding Author: Professor Marisa Karow

Version 0:

Reviewer comments:

Reviewer #1

(Remarks to the Author)

This study addresses a complex and highly relevant question, offering novel insights into the dynamic reactivation of X chromosomes during normal brain development and in neurodevelopmental disorders, as exemplified by Opitz syndrome (OS). This is an important and timely area of research. Overall, this is a very well executed study but we have a few comments that may improve the presentation:

Major comments:

1. The bioinformatics approach appears generally good. However, we find it difficult to assess the robustness of allele identification using SNPs derived from bulk RNA-seq data. Could the authors specify the read length used? It would be very helpful to see the distribution of chrX gene expression before and after applying the >20 reads per allele filter. Why was this particular threshold chosen? I understand that such cutoffs are somewhat arbitrary, but without seeing the distribution, it's challenging to evaluate the validity of the choice.
2. Related to the above comment. How do the authors ensure that the number of reactivated genes identified in A-ctrl is not an artifact of this thresholding strategy? (seems like an outlier in comparison to the other two cell lines) What is the divergence in SNP capture across samples? Could these findings be influenced by technical variables such as sequencing depth? Are the same thresholds applied uniformly across all three cell lines? If so, we believe a relative, or normalized measure (e.g., adjusted for sequencing depth) would be a more robust approach.
3. The observed transcriptional changes in the organoids are very interesting! Did the authors also observe any differences in organoid size during differentiation?
4. A critical assumption underlying the paper is the stability of X-inactivation in iPSCs derived from fibroblasts. This point is mentioned but not directly supported with data. Given that the authors established these cell lines, providing a validation of chrX inactivation stability would strengthen the manuscript.
5. Regarding the mutant clones: I'm missing the sanger sequencing of the mutant clones. The authors demonstrate reduced protein levels and size, but the manuscript is missing to document the specific sequence changes.
6. Could the authors provide a theory or comment on possible explanations on why the discrepancy between their results and prior reports? (suggesting that gene reactivation is dependent on XIST dosage).

Minor comment:

The abbreviation "pm" in the Fig. 4I heatmap is not defined in the figure legend.

Reviewer #2

(Remarks to the Author)

Neurodevelopmental disorders (NDDs) often present differently between sexes. While males may exhibit more severe symptoms during childhood, females frequently experience more pronounced challenges during adolescence and adulthood. In this manuscript, the authors test the hypothesis that dynamic X-chromosome reactivation provides a protective mechanism in female neural tissue, thereby reducing the frequency and severity of NDDs in females.

They explore this hypothesis using a range of experimental models and approaches, addressing a significant knowledge gap regarding the developmental timeline of X-chromosome inactivation (XCI), escape, and the emergence of facultative escape in human cells. By monitoring XCI and escape throughout neural differentiation from a defined point of origin, the authors uncover that many of the reactivating genes are linked to NDDs. Importantly, they demonstrate that differentiation-specific facultative escape is a general phenomenon, not limited to humans.

A particularly novel aspect of the study is the mechanistic insight into the X-linked MID1 gene and its role in Opitz BBB/G syndrome (OS). The experiments are well-designed, and the manuscript is clearly written. I support its publication and offer the following suggestions to further strengthen the conclusions:

Major Points:

1. Could the authors provide a list of all X-linked genes identified in their study that are associated with NDDs and intellectual disability, in addition to MID1? It would also be helpful if they could validate the expression of these genes at the protein level using publicly available protein databases.
2. Regarding Figure 1D, the color coding is unclear. Could the authors clarify which genes are activated in female subjects and which are classified as facultative escapees?
3. The time point chosen for brain organoid analysis (day 30) appears to be relatively early in development. This stage may be too immature to fully assess the impact of MID1 on neuronal identity and maturation. Could the authors justify this choice or consider extending the analysis to later stages?
4. The current data on MID1 are not sufficient to explain the microcephaly and cerebellar hypoplasia observed in OS patients. Additional morphological analyses are needed to determine whether MID1 dysregulation leads to enrichment of microcephaly-associated genes or causes premature death of neural progenitor cells (NPCs).
5. Is MID1 expressed differentially in adult male and female brains? The organoid data suggest that biallelic expression peaks at the progenitor stage and diminishes during differentiation. Does this imply that MID1 is primarily required during the NPC stage rather than for later neuronal specification?
6. It is unclear if MAP2+ neurons have lower- or upper-layer neuronal identity.
7. It remains unclear whether the observed decrease in MAP2 expression affects the physiological or functional properties of neurons. Could the authors provide data on synaptic dynamics. These data would support the genotype–phenotype correlations in human Opitz syndrome.
8. In addition to the observed increase in p21 expression across all three genotypes, it would be interesting to assess whether Protein Phosphatase 2A (PP2A) levels are also elevated, as PP2A has been implicated in the phenotype associated with Opitz syndrome.

Reviewer #3

(Remarks to the Author)

Major comments:

- This is a timely study about a little understood topic, X-chromosome inactivation. This study tries to elucidate the effects of XCI during neural differentiation, while also unravelling the effects of facultative X-reactivation.
- Unfortunately, the findings in this manuscript are not well put together. In an effort to explain as many of the X-linked differentiation phenotypes as possible, the authors have taken a very diffuse approach that is not hypothesis-driven.
- The authors have tried to integrate publicly available data and undertake mouse studies to validate their findings in human iPSCs. Neither approach is well executed, and leaving them out would be recommended.
- The authors seemed to be struggling with how to present their data. The figures are a mess. There are too many figure panels with too much information. A number of figure panels do not actually contribute to the conclusions in the manuscript and can be moved to the supplementary figures. Some figure panels in both main figures and supplementary figures have not been labelled correctly or figure legends do not explain the figure panels accurately.
- The section in organoids is commendable. The breadth of information that can be extracted from this single cell experiment can become the centre-stage of the manuscript. However if the authors feel that the other assays such as QUASEP provide more relevant information, then they should be more judicious about which single cell graphs to add to the main figures. There is a problem with too much information in the main figures that spans the entire manuscript.
- The number of techniques and methods used in this manuscript is commendable, but their implementation does not always add to the central theme. Related to this matter, the titles for the sub-sections in the 'Results' sections should reflect the result, instead of the methods used. For example, "X-reactivation during neural differentiation of human iPSCs" is a good title, but "Analysis of gene expression from the inactive X-chromosome with single cell resolution" is just a description of methods. It may appear that the manuscript was written by multiple authors and hasn't been integrated properly.
- The evidence in support of Opitz BBB/G syndrome (OS) is not well integrated into the manuscript. The clinical symptoms of OS have been mentioned multiple times in the manuscript which is unnecessary. This can be limited to the introduction. The organoid study with OS iPSCs and MID1 iPSCs is very exciting and can be emphasised in the manuscript. Surprisingly, there is no mention of OS in the abstract.

Specific comments:

1. Page 5: "Importantly, reactivated and late-silenced genes showed no significant correlation between overall gene expression levels and allelic expression, confirming that reactivation and late-silencing were not spuriously detected due to lower overall expression in the respective cell types"
- This statement needs further explanation, or validation. Although it is clear there is silencing and reactivation going on, it is

not clear how to interpret Fig1(H).

2. Figures S1 and S2 labelling is confusing. For example, Fig S1B, C, D have figure legends saying 'C' and 'G'. This is repeated in Fig S2. It is not clear what C and G mean in this context.

3. Figure 1B, D, F: Can you point out differentiation stages (iPSC, NPCs, neurons) here please?

4. Page 5: "Further characterization of the different types of biallelically expressed genes revealed that genes of all three classes shared a significantly higher overlap than expected by chance with previously reported biallelically expressed X-chromosomal genes including facultative escapees"

- The method and data source used to undertake this further analysis is not clear. Although there is mention of "44 human tissues" in the next sentence, it is unclear where the tissues are coming from or what they are.

- Authors should add the source of data in the text and reason for using it.

5. Figure S3E: Is this M-ctrl?

- Figure labelling is inconsistent. For example, in Fig S3B, the cell line name is on top and antibodies on the right side. This is switched in Figures S3C onwards.

6. Page 7: "Conservation of dynamic escape in the mouse"

- It is unclear why the authors have added X-inactivation escape study in mouse in an otherwise human study.

- Conservation of interspecies regulatory elements between mouse and human would not be unexpected, so the mouse data does not add anything to the paper other than just validating the human data.

- The authors could try a re-interpretation of this section.

Reviewer #4

(Remarks to the Author)

Version 1:

Reviewer comments:

Reviewer #1

(Remarks to the Author)

The authors have addressed all our comments.

Reviewer #2

(Remarks to the Author)

The authors have thoroughly addressed my concerns and have resubmitted a meticulously revised manuscript. The revisions significantly improve the clarity and scientific rigor of the work. I have no further comments and recommend acceptance.

Reviewer #3

(Remarks to the Author)

Generally, the manuscript reads much better and several of the major and specific comments have been satisfactorily addressed, but some of the structural problems still remain. I recommend the authors to address these.

1. The issue with the findings not well put together, still remains. For example, the use of 2D neurons in the first results section "Reactivation of X-linked genes during neural differentiation" is confusing, especially since there is a big section on brain organoids later. X-inactivation status during differentiation appears to have been measured in brain organoids, so what is the merit of undertaking this in 2D neurons too? 2D neurons are an older model system with no obvious biological advantages compared to brain organoids.

2. The structure of the manuscript still feels confusing. Shouldn't the brain organoid work be presented up front, and mouse work used as validation?

3. From my previous comments about figures, I found fig 4 to be especially busy. For example, can the authors explain figure 4(l)? It is not immediately clear why it has been included in this figure. The caveat of this heatmap is the very small font size for the individual genes which is impossible to read. If the purpose is to highlight certain gene clusters within the heatmap that should be explicitly highlighted on the panel itself and the figure legend should reflect that. If this heatmap is merely informative as suggested by the figure legend it should be in the supplementary figures.

4. Could you add a schematic to explain your findings visually?

Reviewer #4

(Remarks to the Author)

Point-by-point response to reviewers from manuscript NCOMMS-25-40254

We thank the reviewers for thoroughly reviewing our manuscript and providing critical insights and feedback. Based on the recommendations, we have now substantially revised the manuscript and added new data and experiments to support and strengthen our main findings.

We have restructured the manuscript, revised, and edited the text accordingly. Relocated or new paragraphs are highlighted in light grey throughout the manuscript.

Below we provide a summary of the major revisions and changes based on the main and supplementary figures.

Detailed changes:

Figure 1	- addition of data showing an increase in biallelic expression of target genes upon neural differentiation on single cell level. New Figures 1F-K relocated from previous Figure 3.- addition of a graphical scheme exemplifying mono- and biallelic expression in the presence of XIST; new Figure 1E- relocation of previous Figures B, D, F to new Figure S1- relocation of previous Figure 1H to new Figure S1
Figure 2	- relocation of previous Figure 2A to new Figure S3- addition of legend in Figure 2C
Figure 3	- new Figure 3 now dedicated to show biallelic expression in the mouse- previous Figures 3A-H were relocated to the new Figure 1 and the new Figure S3
Figure 4	- unchanged
Figure 5	- unchanged
Figure 6	- unchanged

Figure S1	- Figures S1F-H were relocated from previous Figure 1- Figure S1I was relocated from previous Figure 1- new Figure S1J was added
Figure S2	- unchanged
Figure S3	- Figures S3G, H were relocated from previous Figure 3- Figure S3M was relocated from previous Figure 2
Figure S4	- unchanged
Figure S5	- Figures S5A, B were relocated from previous Figure 6
Figure S6	- new Figure S6 showing the Sanger sequencing results of the genome-edited iPSC lines used in this study.
Figure S7	- Figure S7 derived from previous Figure S6 which was split into two new Figures, now Figures S7 and S8 to increase clarity of data representation
Figure S8	- Figure S8 derived from previous Figure S6 which was split into two new Figures, now Figures S7 and S8 to increase clarity of data representation
Figure S9	- new Figure S9 with newly incorporated data set showing the molecular basis of the gradual brain organoid phenotype depending on the presence or absence of a wildtype MID1 allele on the inactive X-chromosome in the iPSC lines. These data further support the characterization of the phenotype shown in Figures 5 and 6.
Figure S10	- Figure S10 was relabeled from previous Figure S7

Reviewer #1 (Remarks to the Author):

This study addresses a complex and highly relevant question, offering novel insights into the dynamic reactivation of X chromosomes during normal brain development and in neurodevelopmental disorders, as exemplified by Opitz syndrome (OS). This is an important and timely area of research. Overall, this is a very well executed study but we have a few comments that may improve the presentation:

We thank the reviewer for the time spent in reviewing our manuscript and for recognizing the importance of this study. Moreover, we thank the reviewer for the constructive criticism which we address in detail below.

Major comments:

1. The bioinformatics approach appears generally good. However, we find it difficult to assess the robustness of allele identification using SNPs derived from bulk RNA-seq data. Could the authors specify the read length used? It would be very helpful to see the distribution of chrX gene expression before and after applying the >20 reads per allele filter. Why was this particular threshold chosen? I understand that such cutoffs are somewhat arbitrary, but without seeing the distribution, it's challenging to evaluate the validity of the choice.

We fully agree with the reviewer's comment that the robustness of the analysis of allele-specific expression using heterozygous variant sites from the bulk RNA-seq data is essential for the validity of results. Therefore, we paid special attention to ensure the reliability of our methodology. As outlined in the "Allele-specific expression analysis (ASE) of bulk RNA-seq data" method section of our manuscript, we performed multiple stringent quality filtering steps including removing variants with a quality score <100, filtering out multi-allelic variants and long indels (>50 bp), performing statistical analyses only for variant sites with >20 reads with an adjusted p-value threshold of 0.01, and ultimately also performing a manual curation step to remove ambiguous predictions from the analysis. As specified in the "Bulk RNA-sequencing" section, the read length was 76 bp for the M-ctrl and J-ctrl lines and 151 bp for the A-ctrl line, for which sequencing was performed at a later time point. We understand that our selected thresholds for the number of reads per variant site might appear arbitrary, however they were chosen after careful consideration and literature research to balance the trade-off between specificity and sensitivity for the analysis. For instance, previous studies have used a cut-off of 7 (Wainer Katsir and Linial, 2019) or 10 (Tukiainen et al., 2017) reads to select informative SNPs and the prediction of biallelic expression was performed using individual samples. In our analysis, we have 4 replicates for NPCs and neurons and 6-8 replicates for iPSCs in the M-ctrl and J-ctrl line as well as 3 replicates in the A-ctrl cell line. A group level analysis with at least 20 reads per variant site (and > 40 reads in case of 8 replicates) in our opinion increases the statistical rigor while also accounting for biological variability in contrast to previous studies. As the reviewer requested, we also looked at the number of candidate variant sites that showed indication of biallelic expression before and after applying our filter for the number of reads as illustrated in Point-by-point letter Figure 1 (PP Fig.1) below. Importantly, filtering reduced the candidate list by 82% to 93% showing that our filtering is very stringent and conservative (PP Fig. 1a). While we did a hard filtering for variant

quality scores (>100), the filter for the number of reads supporting a variant site led to a further significant increase in the distribution of the quality scores of the variants included in the final statistical analysis (PP Fig. 1b). Interestingly, while the number of individual candidate variant sites varied between cell lines, on the gene level we observed a comparable number of genes including informative variant sites (PP Fig. 1c). Furthermore, even after applying our filtering for the number of reads, we still observed that the candidate variant sites were distributed along the whole X-chromosome (PP Fig. 1d), therefore our analysis was not biased towards a particular region on the X-chromosome.

Point-by-point letter Figure 1. Quality control of candidate variant sites for prediction of biallelic expression. **a**, Bar plot with the number of candidate variant sites before filtering for the number of reads (unfiltered) and after applying the ≥ 20 reads per group filter (filtered) in each cell line. **b**, Box plots with variant quality scores before and after applying the ≥ 20 reads filter. **c**, Bar plot with the number of genes containing candidate variant sites before filtering for the number of reads (unfiltered) and after applying the ≥ 20 reads per group filter (filtered) in each cell line. **d**, Distribution of candidate variant sites along the X-chromosome before and after applying the ≥ 20 reads per group filter. ****, $p < 0.0001$.

2. Related to the above comment. How do the authors ensure that the number of reactivated genes identified in A-ctrl is not an artifact of this thresholding strategy? (seems like an outlier in comparison to the other two cell lines) What is the divergence in SNP capture across samples? Could these findings be influenced by technical variables such as sequencing depth? Are the same thresholds applied uniformly across all three cell lines? If so, we believe a relative, or normalized measure (e.g., adjusted for sequencing depth) would be a more robust approach.

As outlined above, we took multiple precautions to limit the effect of technical variables on our predictions of biallelically expressed genes. However, we acknowledge that the impact of such factors can never be completely excluded from any large-scale statistical analysis. We indeed detected a higher number of biallelic genes in the A-ctrl line. Sequencing of this line was

performed at a later time point and while sequencing depth was comparable with sequencing for the M-ctrl and J-ctrl lines, we increased the read length from 76 to 151 bp with the aim of increasing the sensitivity of SNP calling. Subsequently, the quality filtered table used as input in the statistical analysis for A-ctrl indeed included a higher number of candidate variant sites but a comparable number of genes containing informative variants (see PP Fig. 1). Moreover, we applied the same stringent filtering criteria for all cell lines and the higher number of starting potential candidate sites in A-ctrl is mitigated by the correction for multiple comparisons which becomes more conservative when the data set is bigger. Therefore, we are convinced that the higher number of reactivated genes in the A-ctrl line is not an artifact but reflects the increased sensitivity of the analysis in this cell line. Furthermore, we would like to draw the reviewer's attention to the overlap of biallelic genes detected in multiple cell lines which we highlight in Figure S1 in the manuscript. For instance, *MID1* which takes center stage in the second part of our analysis, is confidently detected as reactivated in all three cell lines. Furthermore, our genes are significantly enriched for previously reported constitutive and facultative escape genes in humans (Figure S3M, Table S2) which strengthens the validity and specificity of our methodological pipeline.

3. The observed transcriptional changes in the organoids are very interesting! Did the authors also observe any differences in organoid size during differentiation?

We thank the reviewer for appreciating the organoid phenotyping. We have not observed any difference in the size of the organoids on day 30. While we are seeing a reduction of MAP2 positive neurons, PAX6 expressing progenitors are enriched in the OS organoids on day 30. At this time point progenitor cells are still an important cellular fraction in the developing organoid. In OS patients, microcephaly is a variable symptom and is diagnosed relatively late during pregnancy (around week 31). Due to the disproportion between MAP2 positive neurons and PAX6 positive progenitor cells we found in the organoids, similar to the OS patients, we expect to see a reduction of organoid size later, when neurons are getting the predominant cell type.

4. A critical assumption underlying the paper is the stability of X-inactivation in iPSCs derived from fibroblasts. This point is mentioned but not directly supported with data. Given that the authors established these cell lines, providing a validation of chrX inactivation stability would strengthen the manuscript.

The reviewer raises a very important point. The long noncoding RNA *XIST* is the master regulator of X-inactivation. It is expressed from the inactive X-chromosome and influences its epigenetic landscape so that gene expression is repressed. Another hallmark of X-inactivation is a territory of H3K27me3 modification found on the inactive X-chromosome. *XIST* expression and detection of H3K27me3 territories indicate stable X-inactivation, while loss of the two marks can mean that X-inactivation is unstable and eroded. We have monitored the X-(in)activation status throughout our experiments by visualizing *XIST* expression and H3K27me3 modification using RNA-FISH (Figures 1F-K, Figures S3B, S3D-L), immunofluorescence (Figures S3B, S3C, S3E) and single cell / single nucleus RNA sequencing (Figure 4). Hence, we show on single cell resolution that *XIST* and H3K27me3 modification is present in the lines used throughout the study. However, in the M-lines (patient lines) and some of the J-lines *XIST* expression was very low. In order to make

sure that the observed reactivation was not a result of unstable X-inactivation, we compared reactivation patterns in *XIST* high and low expressing cells. Similar levels of reactivation were seen in *XIST* high- and low expressing cells throughout the entire manuscript (Figure 4F). In order to show stable inactivation of selected loci in iPSCs we furthermore used allele-specific RT-PCR over many different passages and found reliable inactivation (Figure S8D), which gives additional support that differentiation induced reactivation of gene expression from the inactive X-chromosome is independent of erosion phenomena.

We adapted the paragraphs addressing stability of X-inactivation found across the manuscript (line 162 ff, 377 ff, 514 ff) to clarify and strengthen this point raised by the reviewer.

5. Regarding the mutant clones: I'm missing the sanger sequencing of the mutant clones. The authors demonstrate reduced protein levels and size, but the manuscript is missing to document the specific sequence changes.

We thank the reviewer for pointing this out. We have added the sanger sequencing profiles as a new supplementary figure (Figure S6).

6. Could the authors provide a theory or comment on possible explanations on why the discrepancy between their results and prior reports? (suggesting that gene reactivation is dependent on *XIST* dosage).

We thank the reviewer for this suggestion. It is true, our data suggest that full escape on one hand and reactivation of gene expression from the inactive X-chromosome on the other hand follow a different molecular logic. While for full escapees the expression of alleles from the inactive X-chromosome correlates with *XIST* levels, reactivated genes do not show such a dependency (Figure 4F). The *XIST* dependency of full escapees is in line with previous observations showing that the expression of escapees depends on *XIST* dosage. Similarly, earlier studies showed that erosion of the XCI due to the loss of *XIST* expression results in expression of alleles from the inactive X-chromosome in the iPSCs. The reactivated genes described in our study are monoallelically expressed in iPSCs over many passages (Figure 1B, S8D) and only become biallelically expressed after differentiation and hence describe a different entity than eroding genes. Genes biallelically expressed throughout the whole lineage from iPSC to differentiated cells (full escapees as well as eroded genes) are determined by the establishment of X-inactivation. Reactivation occurs later, when X inactivation is already stably established. Moreover, it has been described that X-inactivation maintenance can be independent of *XIST*.

We think that reactivation of expression from alleles from the X-chromosome follows a distinct mechanism that is induced by specific stimuli, in our case differentiation and likely entails the substantial chromatin reorganization occurring during differentiation. This is similar to recent studies showing reactivation from the inactive X-chromosome due to aging, a process also correlating with reorganization of the chromatin.

We discuss these aspects in the present manuscript on line 514-530.

Minor comment:

The abbreviation “pm” in the Fig. 4I heatmap is not defined in the figure legend.

The reviewer is right. We apologize for this mistake and added the abbreviation (pm = paraxial mesoderm).

Reviewer #2 (Remarks to the Author):

Neurodevelopmental disorders (NDDs) often present differently between sexes. While males may exhibit more severe symptoms during childhood, females frequently experience more pronounced challenges during adolescence and adulthood. In this manuscript, the authors test the hypothesis that dynamic X-chromosome reactivation provides a protective mechanism in female neural tissue, thereby reducing the frequency and severity of NDDs in females. They explore this hypothesis using a range of experimental models and approaches, addressing a significant knowledge gap regarding the developmental timeline of X-chromosome inactivation (XCI), escape, and the emergence of facultative escape in human cells. By monitoring XCI and escape throughout neural differentiation from a defined point of origin, the authors uncover that many of the reactivating genes are linked to NDDs. Importantly, they demonstrate that differentiation-specific facultative escape is a general phenomenon, not limited to humans. A particularly novel aspect of the study is the mechanistic insight into the X-linked MID1 gene and its role in Opitz BBB/G syndrome (OS). The experiments are well-designed, and the manuscript is clearly written. I support its publication and offer the following suggestions to further strengthen the conclusions:

We thank the reviewer for valuable comments and acknowledging the novelty of our data as well as the breadth of our question.

Major Points:

1. Could the authors provide a list of all X-linked genes identified in their study that are associated with NDDs and intellectual disability, in addition to MID1? It would also be helpful if they could validate the expression of these genes at the protein level using publicly available protein databases.

We have reported the association between biallelic genes detected in our study and neurodevelopmental disorders in supplementary Table S2. However, we apologize that we had not explicitly mention this point in the main text of the manuscript. In the revised version the reference to the list asked by the reviewer is mentioned in lines 313-317.

34 out of 95 reactivated genes intersected with high confidence NDD genes as reported in Figure 5A. This overlap was significantly higher than expected by chance. Furthermore, 15 out of the 82

full-escapees and 5 out of the 29 late-silenced genes are also associated with NDDs, however the overlap was not statistically significant (Figure 5A, Table S2).

As suggested by the reviewer, we also examined the protein expression of NDD-associated reactivated genes by employing the Human Protein Atlas (Uhlén et al., 2015). This comprehensive resource includes semi-quantitative data from immunohistochemistry experiments that classify protein levels as not detected, low, medium and high across different tissues. Looking at the brain regions included in the atlas - cerebral cortex, cerebellum, hippocampus and caudate - we confirmed that NDD-associated reactivated genes are expressed in brain tissue at the protein level mostly at high and medium levels (PP Fig. 2).

Point-by-point letter Figure 2. Protein expression of NDD-associated reactivated genes in different brain regions. Expression values were obtained from the Human Protein Atlas.

2. Regarding Figure 1D, the color coding is unclear. Could the authors clarify which genes are activated in female subjects and which are classified as facultative escapees?

We apologize if the color code was misleading in the figure and not described in the figure legend. Since the previous Figure 1D has only one color, we assume that the reviewer potentially refers to Figure 2D? Here, the colors are indicating the reactivated, full escapees (left side of the ideogram), and late silenced genes. We have added a color code legend inside the relocated figure (now Figure 2C) next to the ideogram and apologize for the inconvenience.

3. The time point chosen for brain organoid analysis (day 30) appears to be relatively early in development. This stage may be too immature to fully assess the impact of MID1 on neuronal identity and maturation. Could the authors justify this choice or consider extending the analysis to later stages?

We thank the reviewer for this very important comment. It is true that the d30 timepoint is rather early, yet we do see a considerable fraction of both progenitors and neurons at this stage. In the bulk sequencing approach we saw that a significant proportion of reactivation takes place between iPSCs and NPCs (Figure 1B), so rather early during differentiation. This is supported by the single nucleus data that showed that dual allele usage is more frequent at the progenitor level (SOX2 positive, Figure 4F) and by the re-analysis of the scRNA-seq data of human spinal cord (Figures 4J, S4L). In this analysis we found substantially more reactivation in the early (CS14) embryos compared to the older (CS19) embryos (see Figure 4K). In a pseudotime analysis we

performed with the newly generated scRNA-seq data (new Figure S9), we further see that the impact on the overall transcriptional deviation in the *MID1* mutant samples is highest at the progenitor level (new Figure S9J).

In order to study the role of reactivation of gene expression from the inactive X-chromosome on cellular phenotypes caused by NDD genes, we have therefore focused on the early organoids in this study. In the newly incorporated scRNA-seq data (Figure S9) iPSC lines of all genotypes contribute similarly to the different neuronal lineages present (PP Fig. 3)

We fully agree that in a follow-up study it will be interesting to see how organoids behave when growing older. In this context, very recently, gene reactivation has been described in the aging process (Gadek et al., 2025; Hoelzl et al., 2025).

Point-by-point letter Figure 3. Force-directed graph embedding shows similar contribution of the experimental groups to all lineages.

4. The current data on *MID1* are not sufficient to explain the microcephaly and cerebellar hypoplasia observed in OS patients. Additional morphological analyses are needed to determine whether *MID1* dysregulation leads to enrichment of microcephaly-associated genes or causes premature death of neural progenitor cells (NPCs).

We thank the reviewer for the comment. To address this question, we generated scRNA-seq data from brain organoids of patient and genome-edited lines carrying the C-terminal 4-bp deletion in the X-chromosomal *MID1* gene either hetero- or homozygously. Analysis of this scRNA-seq data set revealed *ZIC1* and *ZIC2* as deregulated genes (Figure S9E) when comparing ctrl and mutant progenitors. Mutations in *ZIC1* and *ZIC2* lead to Dandy-Walker cysts (Grinberg et al., 2004) characterized by malformation and hypoplasia of the cerebellum. Interestingly, similar to *MID1* these genes regulate neuronal differentiation (Aruga et al., 2002). Hence, our data suggest a partial overlap and a surprising molecular crosstalk linking OS and Dandy-Walker pathogenesis. Moreover, compared to wildtype, OS NPCs showed *MID1* genotype dependent (Figure S9H) gradual deregulation of more general cell cycle regulators associated with proliferation and

differentiation (e.g. *CDK6*, *CCND1*, *KIF2A*, *NIN*, *CDC42*, Figure S9H) (Broix et al., 2018; Cappello et al., 2006; Grison et al., 2018; Lange et al., 2009; Shinohara et al., 2013), suggesting that altered cell cycle progression accounts for the accumulation of NPCs at the expense of neurons in OS brain organoids (Figures 5C, D, Figures S7A-D), an observation in line with the BrdU data described in the manuscript (Figure SH-N) and that suggest that *MID1* controls the balance between proliferation and differentiation of NPCs.

These data were added to the revised manuscript.

5. Is *MID1* expressed differentially in adult male and female brains? The organoid data suggest that biallelic expression peaks at the progenitor stage and diminishes during differentiation. Does this imply that *MID1* is primarily required during the NPC stage rather than for later neuronal specification?

Previous prominent studies that have conducted comprehensive investigations on sex biased gene expression in humans based on the GTEx project, have not reported differential expression of *MID1* in adult female and male brains (Lopes-Ramos et al., 2020; Oliva et al., 2020).

The reviewer is right that *MID1* expression is more pronounced in progenitors than neurons (see Figure S9J) where it controls the balance between proliferation and differentiation. The newly incorporated scRNA-seq data reveal a gradual deregulation of more general cell cycle regulators associated with proliferation and differentiation (e.g. *CDK6*, *CCND1*, *KIF2A*, *NIN*, *CDC42*, Figure S9H) (Broix et al., 2018; Cappello et al., 2006; Grison et al., 2018; Lange et al., 2009; Shinohara et al., 2013), suggesting that altered cell cycle progression accounts for the accumulation of NPCs at the expense of neurons in OS brain organoids (Figures 5C, D, Figures S7A-D). Moreover, mutation of *MID1* alters the transcriptional landscape strongest at the progenitor levels (Figure S9J) highlighting the importance of *MID1* early during neuronal differentiation, as suggested by the reviewer.

To follow up on this observation, we examined *MID1* expression in the developing human brain by employing data from the BrainSpan Atlas (<https://www.brainspan.org/>). In agreement with our organoid data, *MID1* expression peaked at earlier developmental stages (post conception week 8-9, 12) and decreased at later stages in both female and male subjects (PP Fig. 4).

Point-by-point letter Figure 4. Heatmap showing *MID1* expression at different stages of embryonic brain development in female and male subjects. Values correspond to log₂ RPKM. Data were obtained from the BrainSpan Atlas (<https://www.brainspan.org/>).

6. It is unclear if MAP2+ neurons have lower- or upper-layer neuronal identity.

We thank the reviewer for raising this important aspect. In fact, the newly incorporated scRNA-seq data of organoids generated from the M-, and J-lines revealed that these lines produce organoids with predominant hindbrain identity (new Figure S9). Therefore, the MAP2+ neurons are not cortical and thus neither of lower nor upper-layer identity. In the new Figure S9 we now show the different lineages present in the organoids, i.e. rhombic lip progenitors (*ATOH1*) giving rise to glutamatergic cells expressing *SLC17A6*, VZ progenitors (*PTF1A*) differentiating into GABAergic cells expressing *GAD2* (Figure S9C).

7. It remains unclear whether the observed decrease in MAP2 expression affects the physiological or functional properties of neurons. Could the authors provide data on synaptic dynamics. These data would support the genotype–phenotype correlations in human Opitz syndrome.

Our organoid tissue phenotyping is based on an image analysis pipeline in which SOX2 and MAP2 pixels are thresholded and counted. We have observed significantly less area covered by MAP2 with a concomitant increase in SOX2 positive areas in OS organoids indicative of less neurons and more progenitors. However, we did not perform a quantification of the protein levels of MAP2 expression within organoid-resident neurons.

To address the question asked by the reviewer we leveraged the newly incorporated scRNA-seq data (Figure S9) to determine scores of gene sets related to neuron maturation (GO:0042551), synapse organization (GO:0050807), and stem cell proliferation (GO:0072089) across samples and along pseudotime (PP Fig. 5). While stem cell proliferation associated gene set showed a

genotype dependent difference early in the lineage as expected by the cellular data described in our manuscript (Figures 5, 6 S7, S10), neither the gene sets associated with neuronal maturation nor synapse organization showed such a genotype dependent deregulation.

Point-by-point letter Figure 5. The scRNAseq dataset from hindbrain organoids was used to calculate expression scores for genes associated with the GO terms neuron maturation (GO:0042551), regulation of synapse organization (GO:0050807), and stem cell proliferation (GO:0072089). The resulting scores were plotted along pseudotime (0=NPC, 1=neurons).

8. In addition to the observed increase in p21 expression across all three genotypes, it would be interesting to assess whether Protein Phosphatase 2A (PP2A) levels are also elevated, as PP2A has been implicated in the phenotype associated with Opitz syndrome.

We have not looked at PP2A levels in the organoids yet. PP2A is highly abundant in all cells of the brain. We had shown that only the microtubule-associated subunit PP2Ac is regulated in a ubiquitin and MID1 dependent manner (Troddenbacher et al., 2001). Quantification of a subcellular fraction of such a highly expressed protein in a selected group of cells on the protein level is very difficult. We will pick this suggestion of the reviewer up in a follow-up study.

However, the radial glia cell marker PAX6, has also been described to be a target of MID1-dependent ubiquitination and degradation (Pfirrmann et al., 2016). Quantification of PAX6 protein levels indeed revealed a substantial increase within the VZLS (ventricular zone like structure) in OS organoids carrying a homozygous loss-of-function mutation in MID1 (J-OS/hom, A-OS/hom) compared to wildtype control. OS organoids carrying such a mutation heterozygous (M-OS/het, J-OS/het, A.OS/het) showed intermediate PAX6 protein levels supporting a gradual rescue by reactivation of MID1 (Figures 6A-D).

Reviewer #3 (Remarks to the Author):

Major comments:

- This is a timely study about a little understood topic, X-chromosome inactivation. This study tries to elucidate the effects of XCI during neural differentiation, while also unravelling the effects of facultative X-reactivation.

We thank the reviewer for the time spent in reviewing our work, considering our work timely, and providing constructive feedback and suggestions.

- Unfortunately, the findings in this manuscript are not well put together. In an effort to explain as many of the X-linked differentiation phenotypes as possible, the authors have taken a very diffuse approach that is not hypothesis-driven.

We appreciate the comment by the reviewer and considered it when revising the manuscript. We have involved a professional editor (<https://www.lifescienceeditors.com>) to assist in restructuring the manuscript so that the reader can more easily understand what has been done.

- (1) We have moved the RNA-FISH experiments from Figure 3 to Figure 1 since this is a validation of the bulk sequencing results, we are describing in Figure 1 on a single cell level.
- (2) This led to Figure 3 solely describing conservation of dynamic allele usage of X-linked genes in the mouse.
- (3) We have added further scRNA-seq data describing the OS cellular phenotype in organoids and showing phenotype rescue through reactivation on a transcriptional level (Figure S9).
- (4) We have divided the result section and have added headings describing the findings rather than technical approaches.
- (5) We have moved parts of Figure 1 (previous Figures 1B, D, F, H), Figure 2 (previous Figure 2A), Figure 3 (previous Figures 3G, H) into a supplementary figure to increase comprehensibility of the main figures.
- (6) We are now mentioning the OS observations in the abstract.
- (7) We have added a description of the OS patient phenotype to the introduction and removed it from the result section.

We think that with these changes the manuscript has improved substantially. We agree with the reviewer that the organoid data and the rescue seen by *MID1* reactivation are the core of the manuscript and think that with the new structure this comes across.

In the first part of our study, we describe our observations on the dynamic differentiation stage-, lineage- and locus-specific allele-usage of X-chromosomal genes, supported by a variety of different techniques. Based on the results from this broader explorative analysis, we then formulated the concrete hypothesis that the reactivation, hence biallelic expression of specific X-chromosomal genes, may have a direct impact on disease development. Subsequently, in the second hypothesis-driven part of our work, we then tested our hypothesis and showed that reactivation of genes from the previously inactive allele can directly result in the amelioration of a disease-phenotype in the context of a neurodevelopmental disorder (Opitz BBB/G syndrome). We apologize for the inconvenience and hope that the changes we have added to the manuscript now better unravels the logic of our study.

- The authors have tried to integrate publicly available data and undertake mouse studies to validate their findings in human iPSCs. Neither approach is well executed, and leaving them out would be recommended.

The main observations in our study were made in *in vitro* systems, raising the question whether the observed developmental stage-, lineage and locus-specific reactivation of gene expression from the inactive X-chromosome also occurs *in vivo*. We therefore consider the mouse data that demonstrate a similar phenomenon *in vivo* in mouse embryos relevant. Likewise, we think that the reanalysis of *in vivo* data from the developing spinal cord of human embryos is very important. These data show that, as in the *in vitro* system, in human embryos allele usage of X-chromosomal genes is highly dynamic. Both systems provide essential and independent evidence for the fact that *in vivo*, besides the previously known escape genes, a substantial additional fraction of genes is dynamically accessed during neural development.

- The authors seemed to be struggling with how to present their data. The figures are a mess. There are too many figure panels with too much information. A number of figure panels do not actually contribute to the conclusions in the manuscript and can be moved to the supplementary figures. Some figure panels in both main figures and supplementary figures have not been labelled correctly or figure legends do not explain the figure panels accurately.

We apologize for potential mistakes in the figure legends which we of course corrected.

On the initiative of the reviewer, we have substantially restructured and revised the manuscript (description see above). We have also involved a professional editor (see above) and think that the manuscript has benefitted substantially.

- The section in organoids is commendable. The breadth of information that can be extracted from this single cell experiment can become the centre-stage of the manuscript. However, if the authors feel that the other assays such as QUASEP provide more relevant information, then they should be more judicious about which single cell graphs to add to the main figures. There is a problem with too much information in the main figures that spans the entire manuscript.

We thank the reviewer for the comment on our organoid data and for appreciating the extent of knowledge gain. In fact, as described above we included a new supplementary Figure (Figure S9) further corroborating the organoid phenotyping data. We have restructured the manuscript and have streamlined the figures based on the advice of the reviewer. The organoid data and the observations we made with the OS organoids are now at the core of the manuscript (see above). We think that the manuscript has improved substantially and thank the reviewer for the valuable advice.

- The number of techniques and methods used in this manuscript is commendable, but their implementation does not always add to the central theme. Related to this matter, the titles for the sub-sections in the 'Results' sections should reflect the result, instead of the methods used. For example, "X-reactivation during neural differentiation of human iPSCs" is a good title, but "Analysis of gene expression from the inactive X-chromosome with single cell resolution" is just a

description of methods. It may appear that the manuscript was written by multiple authors and hasn't been integrated properly.

We are grateful to the reviewer to point out this inconsistency in the titles of the result sections and we changed them accordingly. Now all the titles describe the relevant findings of the respective paragraphs.

- The evidence in support of Opitz BBB/G syndrome (OS) is not well integrated into the manuscript. The clinical symptoms of OS have been mentioned multiple times in the manuscript which is unnecessary. This can be limited to the introduction. The organoid study with OS iPSCs and MID1 iPSCs is very exciting and can be emphasised in the manuscript. Surprisingly, there is no mention of OS in the abstract.

We thank the reviewer for the recognition of the OS data in the manuscript. We fully agree. We have included it in the abstract and streamlined the description of OS throughout the manuscript to avoid redundancy (see above).

Specific comments:

1. Page 5: "Importantly, reactivated and late-silenced genes showed no significant correlation between overall gene expression levels and allelic expression, confirming that reactivation and late-silencing were not spuriously detected due to lower overall expression in the respective cell types"

- This statement needs further explanation, or validation. Although it is clear there is silencing and reactivation going on, it is not clear how to interpret Fig1(H).

We appreciate that the reviewer recognizes the occurrence of reactivation and late-silencing in our experimental set-up. However, one of our main priorities in the analysis was to ensure that our predictions are robust and not due to technical artifacts. Therefore, it is crucial to demonstrate that the lack of biallelic expression in iPSCs for reactivated genes, for example, is not attributable to an overall absence of gene expression in this cell type. The correlation plot in (now relocated to Figure S1H; previous Figure 1H) shows that our estimates for biallelic expression are in fact not dependent on overall gene expression for reactivated and late-silenced genes. To further clarify this point, we here provide an example plot with overall expression levels and biallelic expression across differentiation stages for *MID1* and have added this to the manuscript (Figure S1J and PP Fig. 6) below). This representation clearly demonstrates that *MID1* is stably expressed in all cell types, however, biallelic expression is detectable only in NPCs and/or neurons. As it is not feasible to show individual plots for all genes, the correlation plot in Figure S1H is a compact representation summarizing these results. We have updated the manuscript text to convey this point more clearly.

Point-by-point letter Figure 6. Overall gene expression (left) and biallelic expression (right) for *MID1* at different differentiation stages in the three cell lines A, M and J. Overall expression is shown as log₂ transcripts per million (TPM) counts and biallelic expression corresponds to the allelic ratio of reads mapping to Xi to the sum of reads mapping to Xi and Xa. Xi: inactive X-chromosome, Xa: active X-chromosome.

2. Figures S1 and S2 labelling is confusing. For example, Fig S1B, C, D have figure legends saying 'C' and 'G'. This is repeated in Fig S2. It is not clear what C and G mean in this context.

We apologize that this labelling was not intuitive. The QUASEP methodology relies on heterozygous variant sites (e.g. single nucleotide polymorphisms) to quantify allele specific expression. The small boxes with the letter therefore indicate the DNA base at each of the two alleles for the respective heterozygous variant site. The plot then shows the relative distribution of the expression of either one or the other allele as revealed via QUASEP. We added this information to the respective figure legend.

3. Figure 1B, D, F: Can you point out differentiation stages (iPSC, NPCs, neurons) here please?

The UpSet plots shown in previous Figures 1B, D and F have been moved to Figure S1 in accordance with the reviewer's suggestion to optimize the main figures. In the plots we show the quantification of the number of biallelically expressed genes in each category and respective cell line as well as overlaps between cell lines. Here each type of biallelically expressed gene is associated with a particular differentiation stage: reactivated genes are monoallelic in iPSCs and become biallelic in NPCs/neurons, full escapees are biallelic at all differentiation stages (iPSCs, NPCs and neurons) and late-silenced genes are biallelic in iPSCs and become monoallelic in NPCs and neurons. We hope this description clarifies the results.

4. Page 5: "Further characterization of the different types of biallelically expressed genes revealed that genes of all three classes shared a significantly higher overlap than expected by chance with previously reported biallelically expressed X-chromosomal genes including facultative escapees"

- The method and data source used to undertake this further analysis is not clear. Although there is mention of “44 human tissues” in the next sentence, it is unclear where the tissues are coming from or what they are.

In an effort to not overcrowd the results, we included a very detailed description of how each analysis was performed and which data set was used only in the Methods section. The methodology for the analysis of the overlap with previously reported escape genes with our results, for example, is described in the section “Overlap with known escape genes in humans” in the Methods. However, we acknowledge that this might make it difficult for the reader to follow our motivation for conducting certain analyses while reading the main text. We have therefore updated the main text to include more details on source data and methodology and we hope that our results are now clearer and more understandable.

- Authors should add the source of data in the text and reason for using it.

We have provided detailed information on the source data for each analysis in the Methods section of the manuscript. To improve the clarity of the manuscript, we have now followed the reviewer’s suggestion and have included further details on the source data and our motivation for integrating different data sets in the main text of the result section.

5. Figure S3E: Is this M-ctrl?

Figure S3E shows results for the hoik1 cell line which is another *XIST* high expressing control line. The information of all lines we used in our study is summarized in Table S1. We apologize if this was not clearly pointed out.

- Figure labelling is inconsistent. For example, in Fig S3B, the cell line name is on top and antibodies on the right side. This is switched in Figures S3C onwards.

We thank the reviewer for their attention to detail. We adjusted the labeling to be consistent within each figure and to increase clarity.

6. Page 7: “Conservation of dynamic escape in the mouse”

- It is unclear why the authors have added X-inactivation escape study in mouse in an otherwise human study.

The main observations in our study were made in *in vitro* systems, which raises the question if the observed developmental-stage-, lineage and locus-specific reactivation of gene expression from the inactive X-chromosome can be observed *in vivo*. We therefore consider the mouse data that demonstrate a similar phenomenon *in vivo* in mouse embryos relevant. We changed the figure layout (solely mouse data in the new Figure 3 now) and adapted the result text to better convey this message.

- Conservation of interspecies regulatory elements between mouse and human would not be

unexpected, so the mouse data does not add anything to the paper other than just validating the human data.

- The authors could try a re-interpretation of this section.

Facultative escape is less well described in mouse compared to human. Some substantial differences in the dosage compensation and the inactivation of one X-chromosome between the two species have been described. We think that with the reactivation of X-chromosomal genes we describe an important phenomenon during neural differentiation. We are using the mouse data to show that, even if they display some differences in XCI compared to humans, dynamic escape from X-inactivation exists in the murine nervous system. As suggested by the reviewer we have re-phrased and re-interpreted parts of the respective section.

Reviewer #4 (Remarks to the Author):

We thank reviewer 4 for the time spent to review our work and for providing suggestions to further improve our manuscript.

References:

- Aruga, J., Inoue, T., Hoshino, J. and Mikoshiba, K.** (2002). Zic2 controls cerebellar development in cooperation with Zic1. *J Neurosci* **22**, 218–25.
- Broix, L., Asselin, L., Silva, C. G., Ivanova, E. L., Tilly, P., Gilet, J. G., Lebrun, N., Jagline, H., Muraca, G., Saillour, Y., et al.** (2018). Ciliogenesis and cell cycle alterations contribute to KIF2A-related malformations of cortical development. *Hum Mol Genet* **27**, 224–238.
- Cappello, S., Attardo, A., Wu, X., Iwasato, T., Itohara, S., Wilsch-Bräuninger, M., Eilken, H. M., Rieger, M. A., Schroeder, T. T., Huttner, W. B., et al.** (2006). The Rho-GTPase cdc42 regulates neural progenitor fate at the apical surface. *Nat Neurosci* **9**, 1099–107.
- Gadek, M., Shaw, C. K., Abdulai-Saiku, S., Saloner, R., Marino, F., Wang, D., Bonham, L. W., Yokoyama, J. S., Panning, B., Benayoun, B. A., et al.** (2025). Aging activates escape of the silent X chromosome in the female mouse hippocampus. *Sci Adv* **11**, .
- Grinberg, I., Northrup, H., Ardinger, H., Prasad, C., Dobyns, W. B. and Millen, K. J.** (2004). Heterozygous deletion of the linked genes ZIC1 and ZIC4 is involved in Dandy-Walker malformation. *Nat Genet* **36**, 1053–5.
- Grison, A., Gaiser, C., Bieder, A., Baranek, C. and Atanasoski, S.** (2018). Ablation of cdk4 and cdk6 affects proliferation of basal progenitor cells in the developing dorsal and ventral forebrain. *Dev Neurobiol* **78**, 660–670.
- Hoelzl, S., Hasenbein, T. P., Engelhardt, S. and Andergassen, D.** (2025). Aging promotes reactivation of the Barr body at distal chromosome regions. *Nat Aging* **5**, 984–996.

- Lange, C., Huttner, W. B. and Calegari, F.** (2009). Cdk4/cyclinD1 overexpression in neural stem cells shortens G1, delays neurogenesis, and promotes the generation and expansion of basal progenitors. *Cell Stem Cell* **5**, 320–31.
- Lopes-Ramos, C. M., Chen, C.-Y., Kuijjer, M. L., Paulson, J. N., Sonawane, A. R., Fagny, M., Platig, J., Glass, K., Quackenbush, J. and DeMeo, D. L.** (2020). Sex Differences in Gene Expression and Regulatory Networks across 29 Human Tissues. *Cell Rep* **31**, 107795.
- Oliva, M., Muñoz-Aguirre, M., Kim-Hellmuth, S., Wucher, V., Gewirtz, A. D. H., Cotter, D. J., Parsana, P., Kasela, S., Balliu, B., Viñuela, A., et al.** (2020). The impact of sex on gene expression across human tissues. *Science* **369**,.
- Pfirrmann, T., Jandt, E., Ranft, S., Lokapally, A., Neuhaus, H., Perron, M. and Hollemann, T.** (2016). Hedgehog-dependent E3-ligase Midline1 regulates ubiquitin-mediated proteasomal degradation of Pax6 during visual system development. *Proc Natl Acad Sci U S A* **113**, 10103–10108.
- Shinohara, H., Sakayori, N., Takahashi, M. and Osumi, N.** (2013). Ninein is essential for the maintenance of the cortical progenitor character by anchoring the centrosome to microtubules. *Biol Open* **2**, 739–49.
- Trockenbacher, A., Suckow, V., Foerster, J., Winter, J., Krauss, S., Ropers, H. H., Schneider, R. and Schweiger, S.** (2001). MID1, mutated in Opitz syndrome, encodes an ubiquitin ligase that targets phosphatase 2A for degradation. *Nat Genet* **29**, 287–94.
- Tukiainen, T., Villani, A.-C., Yen, A., Rivas, M. A., Marshall, J. L., Satija, R., Aguirre, M., Gauthier, L., Fleharty, M., Kirby, A., et al.** (2017). Landscape of X chromosome inactivation across human tissues. *Nature* **550**, 244–248.
- Uhlén, M., Fagerberg, L., Hallström, B. M., Lindskog, C., Oksvold, P., Mardinoglu, A., Sivertsson, Å., Kampf, C., Sjöstedt, E., Asplund, A., et al.** (2015). Tissue-based map of the human proteome. *Science (1979)* **347**,.
- Wainer Katsir, K. and Linial, M.** (2019). Human genes escaping X-inactivation revealed by single cell expression data. *BMC Genomics* **20**, 201.

Point-by-point response to reviewers from manuscript NCOMMS-25-40254

We thank the reviewers for thoroughly reviewing our manuscript and providing critical insights and feedback. Based on the recommendations, we have revised the manuscript and highlighted changes in the manuscript in grey.

Reviewer #1

The authors have addressed all our comments.

We thank the reviewer for the effort reviewing our manuscript.

Reviewer #2

The authors have thoroughly addressed my concerns and have resubmitted a meticulously revised manuscript. The revisions significantly improve the clarity and scientific rigor of the work. I have no further comments and recommend acceptance.

We thank the reviewer for the time and effort devoted to evaluating our manuscript, and for expressing appreciation for the improvements incorporated in the revised version.

Reviewer #3

Generally, the manuscript reads much better and several of the major and specific comments have been satisfactorily addressed, but some of the structural problems still remain. I recommend the authors to address these.

We thank the reviewer for the time and effort in evaluating our manuscript, and for acknowledging the improvements made in the revised version.

1. The issue with the findings not well put together, still remains. For example, the use of 2D neurons in the first results section “Reactivation of X-linked genes during neural differentiation” is confusing, especially since there is a big section on brain organoids later. X-inactivation status during differentiation appears to have been measured in brain organoids, so what is the merit of undertaking this in 2D neurons too? 2D neurons are an older model system with no obvious biological advantages compared to brain organoids.

We appreciate the reviewer’s suggestion, and we agree that every model system has its advantages and disadvantages. While brain organoids are a more elaborate system providing a tissue-like context to model early developmental process they are composed of a plethora of different cell types (neural stem cells, neurons, astrocytes, choroid plexus cells, neural crest cells, melanocytes). The 2D differentiation to NPCs and neurons provides a much more homogenous system in which the vast majority of the cells are either NPCs or neurons. The huge technical advantage of such a homogenous system is that we can use bulk approaches to measure biallelic expression, such as bulk RNA-seq and QUASEP. In particular the bulk RNA-seq allows to describe the transcriptome of a system much deeper which is particularly important for the detection of expression from the inactive X-chromosome because those transcripts are significantly less abundant than the ones from the active X-chromosome. A further big advantage of the 2D differentiation approach is that we can use pre-mRNA FISH to detect biallelic expression and correlated it with *XIST* expression on a single cell level. Importantly, this is a completely independent method to validate the findings from bulk RNA-seq and single cell RNA-seq in organoids and human in vivo tissue. In sum, while providing a less elaborate biological model the 2D NPC and neuron differentiation system is more amenable to different techniques (bulk RNA-seq, QUASEP, pre-mRNA FISH) rendering the 2D

system crucial for the description and validation of our findings. To make this argumentation more accessible to the reader we adapted the manuscript text accordingly.

2. The structure of the manuscript still feels confusing. Shouldn't the brain organoid work be presented up front, and mouse work used as validation?

We appreciate the reviewer's suggestion. The logic in the manuscript is to first describe X-reactivation as a new concept in a more simple human model system (2D). In a second step we show that this new concept is evolutionary conserved between humans and mouse. We then dissect in more detail how this new concept of reactivation of X-chromosomal genes differs between cell types and differentiation lineages in organoids and *in vivo* in the developing human spinal cord. In the last section we show how the reactivation of X-chromosomal genes contributes to the development of disease phenotypes. Thus, the mouse data do not serve solely as a validation of the organoid experiments. Rather, they show that, despite fundamental differences in X-inactivation mechanisms between mouse and human, reactivation of gene expression from the inactive X-chromosome is a conserved phenomenon across species. We adapted the manuscript to highlight this more.

3. From my previous comments about figures, I found fig 4 to be especially busy. For example, can the authors explain figure 4(l)? It is not immediately clear why it has been included in this figure. The caveat of this heatmap is the very small font size for the individual genes which is impossible to read. If the purpose is to highlight certain gene clusters within the heatmap that should be explicitly highlighted on the panel itself and the figure legend should reflect that. If this heatmap is merely informative as suggested by the figure legend it should be in the supplementary figures.

We appreciate the reviewer's feedback regarding the complexity of the data in Figure 4, and acknowledge that the font size of the gene names in the heatmap (Fig. 4l) is quite small. The heatmap demonstrates lineage-specific biallelic expression of X-chromosomal genes, with significant variation across different lineages. This differentiation lineage-specific reactivation of X-chromosomal genes is a central finding of our work. To improve readability, we increased the font size indicating the lineages. Furthermore, we have included a new supplementary table (Table S4) listing all gene names (with the quantitative data showing the biallelic expression) in addition to the figure. Moreover, we added a hierarchical clustering to the figure to highlight gene groups that behave similar across different lineages as suggested by the reviewer.

The upper section of Figure 4 (panels A-F) illustrates the snRNA-seq data generated from brain organoids, while the lower section (panels G-K) displays data from the published human *in vivo* spinal cord scRNA-seq dataset. This creates a natural division between the two datasets. Nevertheless, we believe it is important to present both datasets within a single figure, as they collectively reveal the extent of biallelic expression in human tissue-like samples (organoids) and *in vivo* tissues. We have also refined the figure legend to further improve clarity and comprehensibility.

4. Could you add a schematic to explain you findings visually?

We thank the reviewer for this excellent suggestion. In response, we have added two new panels to our figures to improve clarity in presenting our findings. Figure 1L now provides a summary of the results obtained from our 2D system. Additionally, we have replaced the previous Figure 6E with an updated version that summarizes the key conclusions drawn from the patient-specific organoids, as well as the broader implications of our observations

Reviewer #4

I co-reviewed this manuscript with one of the reviewers who provided the listed reports. This

is part of the Nature Communications initiative to facilitate training in peer review and to provide appropriate recognition for Early Career Researchers who co-review manuscripts.

We thank the reviewer for the time spent in reviewing our manuscript.